

# LABS: an agent-based run-out program for shallow landslides

Richard Guthrie[1] and Andrew Befus[2]

[1]Director Geohazards and Geomorphology, Stantec, 200-325 25 St SE, Calgary, T2A 7H8, Alberta, Canada.
Tel.: 4034415133
[2]Software Developer, Environmental Services, Stantec, 200-325 25 St SE, Calgary, T2A 7H8, Alberta, Canada.
Tel.: 4037814116

*Correspondence to*: Richard Guthrie (richard.guthrie@stantec.com)

**Abstract.** Credible models of landslide runout are a critical component of hazard and risk analysis in the mountainous regions worldwide. Hazard analysis benefits enormously from the number of available landslide runout models that can recreate events and provide key insights into the nature of landsliding phenomena. Regional models that are easily employed, however, remain

a rarity. For debris flows and debris avalanches, where the impacts may occur some distance from the source, there remains a need for a practical, predictive model that can be applied at the regional scale. We present, herein, an agent-based simulation for debris flows and debris avalanches called LABS. A fully predictive model, LABS employs autonomous sub-routines, or agents, that act on an underlying DEM using a set of probabilistic rules for scour, deposition, path selection, and spreading behavior. Relying on observations of aggregate debris flow behavior, LABS predicts landslide runout, area, volume, and depth along the

landslide path. The results can be analyzed within the program, or exported in a variety of useful formats for further analysis. A key feature of LABS is that it requires minimal input data, relying primarily on a 5 m DEM and user defined initiation zones, and yet appears to produce realistic results. We demonstrate the applicability of LABS using two very different case studies from distinct geologic, geomorphic, and climatic settings. The first case study considers sediment production from the steep slopes of Papua, the island province of Indonesia; the second considers landslide runout as it affects a community on Vancouver

Island off the west coast of Canada. We show how LABS works, how it performs compared to real world examples, what kinds of problems it can solve, and how the outputs compare to historical studies. Finally, we discuss its limitations and its intended use as a predictive regional landslide runout tool. LABS is freely available to not for profit groups including universities, NGOs and government organizations.

## 1 Introduction

Mountains occupy 30.5% of the global land surface (Sayre et al., 2018), provide much of the global water supply, critical economic resources, and directly support hundreds of millions of people around the world. Steep rugged mountain slopes, however, are also responsible for some of the world's deadliest hazards, threatening infrastructure and causing the loss of thousands of lives annually (on average) (Froude and Petley, 2018).

E-mail: Andrew Befus (andrew.befus@stantec.com)





Debris flows and debris avalanches are potentially destructive, rapid to extremely rapid landslides that tend to travel considerable distance from their source. Interaction between debris flows and objects, resources, or people at distal points along their travel path results in a potentially unexpected and dangerous mountain hazard. One of the critical challenges to overcome with respect to debris flow hazards is, therefore, the credible prediction of size, runout, and depth.

Debris flow runout behavior is controlled by topography, geology (surficial and bedrock), rheology, land use, land cover, water content, and landslide volume. Modeling approaches for predicting debris flow runout have included empirical methods such as total travel distance (Corominas, 1996; VanDine, 1996) or limiting criteria (Iverson, 1997; Benda and Cundy, 1990; Berti and Simoni, 2014), volume balance methods (Fannin and Wise, 2001; Guthrie et al., 2010), analytic solutions and continuum-based dynamic models (Hungr, 1995; O'Brien et al., 1993; McDougall and Hungr, 2003; Rickenmann, 1990; Gre-

goretti et al., 2016; Hussin, 2011), and cellular automata (Guthrie et al., 2008; Tiranti et al., 2018; Deangeli, 1995; D'Agostino et al., 2003).

A limited number of models have been applied regionally (Chiang et al., 2012; Guthrie et al., 2008; Horton et al., 2013) in part due to the complexity of data inputs. Analytical models in particular, while producing excellent results, are frequently complex and can require back analyses to determine model parameters. Hussin (2011), for example, successfully recreated a

15 channelized debris flow in the Southern French Alps, but also found that the model results were sensitive to small changes in the entrainment coefficient, turbulent coefficient, friction coefficient, and the DEM itself. Adjustments to model parameters can require considerable expertise and complicate the predictive value of the models if applied regionally.

There remains a need for a widely accessible debris flow model that produces credible results with limited inputs.

Guthrie et al. (2008) created a regional landslide model intended to provide evidence that the occurrence of the rollover

effect in landslide magnitude frequency distributions was primarily a result of landscape dynamics rather than data censoring (or other causes). That model used cellular automata methods wherein individual cells (agents) followed simple rules for scour, deposition, path selection, and landslide spread. The model assumed aggregate behavior of rapid or extremely rapid flow-type landslides based on about 1,700 data points from Coastal British Columbia (Guthrie et al., 2008, 2010; Wise, 1997). Landslide behavior relied on empirical observations that events exhibit similar scour, deposition, depths, and runout independent of geol-

ogy, rheology, triggering mechanisms, or antecedent conditions. Simply put, once triggered, debris flows and debris avalanches had behavior that tended to be broadly self-similar. The model itself did a credible job of reproducing landslides across a broad region using limited inputs.

The current authors identified a use case and designed, from scratch using C+ and XAML, the landslide runout model presented herein. LABS (Landslides: Agent Based Simulation) is a standalone agent-based model that requires limited inputs and

30 provides the user with both visualization and analytic capabilities. LABS is freely available to non-profit groups (universities, NGOs, government). LABS may be downloaded here **[ADDRESS TO BE PROVIDED IN FINAL SUBMISSION]**.

This paper explains the basis for LABS and provides two very different case studies to demonstrate how it might be applied.



## 2 Description of the Program

LABS estimates sediment volume (erosion and deposition) along a landslide path by deploying '*agents*', or *autonomous sub-routines* over a 5 m spatial resolution digital elevation model (DEM). The DEM surface provides basic information to each agent, in each time-step, that triggers the rule set that comprises the subroutine. In this manner, agents interact with the surface
and with other agents. Each agent occupies a single pixel in each time step.

### 2.1 Agent Generation

The user defines a starting location by injecting a single agent (5 m x 5 m), a group of nine agents (15 m x 15 m initiation zone), or by painting a user defined initiation zone (unlimited size) as indicated by field morphology. Multiple agents may be generated at the same time using any of these methods, or any combination of these methods. LABS can automatically create
15 m x 15 m initiation zones (nine agents) for each point in an imported point file.

The starting location of a single agent, or a group of connected agents, represents the initiation of a landslide. Each landslide *knows* which agents that belong to it, whatever the method of initiation.

### 2.2 Agent Mass

Agents follow probabilistic rules for scour (erosion) and deposition at each time step based on the underlying slope. Rules for
scour and deposition are independent probability distributions for 12 slope classes (bins), modified from Guthrie et al. (2008) to account for a wider range of slopes than the original study (Table 1). They are based on data gathered for coastal BC by Wise (1997) and by Guthrie et al. (2008, 2010), and the results are inferred to be representative of aggregate debris flow behavior elsewhere.

Additional rules for deposition are implemented when agents change cardinal direction. This is a user defined parameter
provided as a substitute for frictional deposition. In this study, deposition was limited to 5% of agent volume for each 45$^o$ turn, but users can experiment with increased amounts.

In each time step an agent scours, deposits, then checks its mass balance. Mass balance is recorded by the agent in each time step, and agents are terminated when their mass equals zero.

### 25 2.3 Agent Path Selection

Agents with mass move down slope in successive time steps by calculating the elevations of the Moore neighbors (the surrounding eight squares in a grid), determining the lowest three pixels and moving to the lowest unoccupied pixel of the three (Figure 1). Should the lowest three pixels be occupied, or should some of the pixels be equal elevation, the agent will merge with one of the cells based on similar internal decision-making rules.



## 2.4   Agent Spread

LABS software has functions that control the spreading behavior of landslides. Spreading behavior causes landslides to re-distribute mass by generating new agents and transferring mass to them from existing agents. Spreading, and therefore the spawning of new agents, is based on the probability density function using a Gaussian normal distribution over the surface (a
5 3D bell curve) where the shape of the Gaussian surface is controlled by the user (Figure 2).

User controls include a fanning slope limit, above which agents will not spawn, and shape controls that determine how steep and narrow the curve, or alternatively how low and broad the curve, for both steeper and flatter slopes.

Spread is calibrated experimentally based on empirical or observed behaviors of actual landslides. In the absence of observ-
10 able landslides, the authors recommend using 27º.

Spread behavior produces realistic results related to underlying topography such that mass is redistributed at sudden changes in slope (e.g. Figure 3), or through gradual slope change where landslides tend to widen and deposit.

Spawned agents immediately perform the same rules as other agents in each time step (including the time step in which they
were spawned).

## 2.5   Agent Tracking

Each agent leaves behind a track that identifies the changes to the pixel, and colors the track according to those changes. This track also provides the visual cue that shows the landslide path.

## 2.6   Outputs

LABS produces results from: a single run, single landslide; single run, multiple landslides; multiple runs, single landslide; or multiple runs, multiple landslides. Following a run or set of runs, each pixel can be queried to provide information about the debris depth (net deposition) at that location, the landslide number, the pixel facing direction, and basic topographic information such as elevation and location. Multiple runs also provide some basic statistics including the number of times a pixel was occupied by an agent, and the minimum and maximum debris depth over all runs.
Each pixel is colored to represent scour or deposition. Red through yellow represent net scour (red being deeper than yellow), and green through blue represent net deposition (blue being deeper than green). Grey colors represent no change, or in the case of multiple runs, they represent no average change in depth (transition zones).

Figure 4 demonstrates the difference in scour and deposition along a landslide path. The reader can see that road(s) tend to accumulate sediment, consistent with observations on Vancouver Island (Guthrie et al., 2010). Similarly, scour on the fill slope
side of the road, where it is locally steeper, is also easily observed.



Figure 5 shows an example of LABS output within the program, in this case, for multiple landslides. Once again, colors relate to erosion and deposition.

Landslide predictions are probabilistic and thus vary between runs. Individual landslides can be shorter than those observed in the field in any given run, but the runout over multiple runs (e.g. 50) tends to be longer than those mapped since it includes more iterations of landslides. Longer landslides should not be confused with debris floods; LABS is based on observations of erosion and deposition in debris flows or debris avalanches, and the aggregate behavior of debris floods are based on substantially different mechanics not captured in the model.

### 2.6.1 Export to Excel

Landslide specific information (landslide number, area, volume) can be exported as an Excel file. The output allows the user to analyze magnitude frequency characteristics of the modeled landslides, including area and volume from the entire footprint, the erosion, and deposition zones, and confirm credible results.

### 2.6.2   Export to Shapefile

Data is easily exported to a shapefile through either an export points function, or an export to layer function. The first converts each pixel and associated metadata for each landslide to a point file for analysis in GIS software, while the second exports the metadata to an existing shapefile allowing, for example, the user to estimate cumulative sediment contribution to previously mapped polygons.

### 2.6.3   Export to Geotiff

LABS exports the modeled landslides as geotiffs to enable viewing in other software and visual comparison with existing ground conditions.

### 3 Model Performance

LABS predicts landslide paths well, it spreads and converges in a realistic manner, and it credibly predicts runout, erosion, and deposition. When compared to actual landslides, LABS tends to produce meaningful results (Figure 6).

Due to the probabilistic nature of the model, individual landslide runs produce varying results and actual landslides may occupy a different footprint and have longer or shorter runout than an equivalent modeled landslide. For the same reason, the





cumulative footprint of multiple runs (e.g. 50) will tend to show longer runout or occupy more ground than individual mapped landslides.

Distal margins of landslides in multiple runs tend to be inundated less frequently than the main landslide body. Interpretation of the cumulative footprint should consider both the range of deposition or erosion, and the number of times that the pixel was
5 occupied.

Landslide area volume relationships compared favorably to historical studies (Figure 7 and Table 2) corroborating that, once again, the model produces reasonable results.

Running LABS with multiple landslides in any given run tends to result in increased scour or deposition wherever those landslides converge and interact, as compared to an individual event.

## 3.1 Case Studies

To better understand how LABS performs and how it might be applied, additional results are described in each of two unique case studies below.

## 3.2 Case Study I: Debris Flows in Papua, Indonesia

### 3.2.1 Background

Tembagapura is a high alpine town, 2000 m above sea level, in the Jayawijaya Mountains in the Mimika Regency of the Province of Papua, Indonesia (Figure 8). Formed from uplifted and accreted terrains driven by the oblique convergence of the Pacific and Indo-Australian plates (Davies, 2012) Tembagapura is entirely surrounded by steep mountain slopes that regularly
produce landslides including debris flows and debris floods.

In 2017, debris floods swept through the town causing considerable damage, and town authorities sought to better understand the expected magnitude and frequency of debris floods to better mitigate and prepare for future events.

A landslide inventory was conducted using remotely acquired vertical color imagery from 2012, 2016, and 2017. The in-
25 ventory resulted in 375 mapped landslides (Figure 9) in the Tembagapura watershed and revealed that landslide evidence had a short persistence time in the dense and verdant vegetation (see Guthrie and Evans (2007) for a discussion of geomorphic persistence).

Rapid weathering and soil formation was inferred to provide a near infinite sediment supply that moves through the watershed in a "conveyor-belt" type process, whereby weathered rock was transported to the river system and subsequently transported
downstream in successively larger floods.





With multiple landslides occurring annually, a relationship describing landslide triggering rainfall was built from terrain mapping, the landslide inventory, weather data, and the town records of landslide causing precipitation events (Figure 10). In order to supply a debris flood model, the amount of sediment generated by landslides, and thus contributing to the conveyor
belt of available sediment, was modeled in LABS.

### 3.2.2  Calibrating the Model

In order to calibrate LABS, landslide head scarps were painted onto a 5 m DEM using the *Paint Agents* tool in LABS and by looking at the imported existing inventory shapefile. LABS has several methods of creating landslide initiation zones including
*Inject Agents*, a tool that inserts agents one 5 m pixel at a time, *Paint Agents*, a tool that allows the user to paint the imagined initiation zone as large as they want, but no less than 15 m x 15 m (nine agents), and a *Create Agents From Points* tool that automatically creates a 15 m x 15 m (nine agent) grid at user defined, or imported, points. In this case, landslide initiation zones were estimated visually on the LiDAR image (Figure 11).

The landslide simulation was activated (toggle the *Go* button) and the results compared to mapped landslides and on-the-ground results (Figure 5,Figure 6, Figure 12).

The probabilistic nature of the model meant that every run was slightly different, but LABS produced morphologically meaningful results when comparing flow paths, scour and deposition regions, divergence, convergence and runout. In addition,
the magnitude frequency statistics between mapped and modeled landslides plotted similarly (Figure 13) and volumes, when compared with previous experience Table 2 were deemed credible.

### 3.2.3  Calculating Sediment Production

Six significant debris flood producing storms since 1991 were identified based on town records (1998, 2010, 2013, 2014, 2016,
and 2017). In order to simulate sediment made available to the conveyor belt system of sediment production during major storms, landslides were generated from random points in the slopes above Tembagapura (Figure 14 and Figure 15), stratified by susceptibility (part of the terrain mapping exercise) and based on the storm return period and the relationship from Figure 10. The results (scour and deposition) were accumulated into existing terrain polygons for the periods between each significant debris flood



Sediment accumulated between storms was predicated on an assumed average landslide frequency (Figure 10) for return periods of one to three years. Collectively, this produced an estimate of the slope-based sediment contribution to historical landslide events. Landslide generated sediment from the LABS model was exported to shapefiles and reported for each of 11 sub-basins that formed the watersheds above Tembagapura (Table 3 ; sub-basins are visible in Figure 14 and Figure 15 as pink lines). A debris flood model was then run through the system using the accumulated sediment as a bulking factor and compared to actual events.

Once the historical events were calibrated, design floods were determined by bulking the debris flood model with sediment estimated for specific storm return periods (Table 4). In this case sediment was accumulated in the model under the assumption that no debris flood greater than the 5-year event had occurred in the preceding 5 years. The results allowed the user to estimate hydrograph bulking for the 10, 20, 25, 50, and 100 year events.

### 3.2.4 Case Study I Summary

LABS successfully simulated debris flows in the steep mountains surrounding Tembagapura. Scour zones were painted using the supplied tool, and for predictive analysis were created automatically from randomly generated points using an imported shapefile.

The results were used to estimate landslide generated sediment to the stream network subsequently flushed by periodic debris floods. The model produced morphologically meaningful results and similar magnitude frequency characteristics to mapped landslides. The sediment contribution from slopes was easily exported to shapefiles for analysis and summation and ultimately to provide volume estimates for hydrograph bulking in the debris flood model at user specified design floods.

### 3.3   Case Study II: Understanding of Risk from Debris Flows on Vancouver Island

### 3.3.1   Background

Vancouver Island comprises approximately 31,788 km$^2$ of rugged terrain between sea level and 2,200 m elevation off the Canadian west coast. Oriented NW-SE, the steep Vancouver Island Ranges form the volcanic backbone of the island. Basalt and andesite are intermixed with marine sedimentary rocks, intruded in turn by granitic batholiths (Yorath and Nasmith, 1995). Pleistocene glaciation carved deep fjords and inlets, and created over-steepened U-shaped valleys that characterize the topography today. Precipitation varies between 700 mm$^{-1}$ and over 6,000 mm$^{-1}$ and landslides are common with rates between





0.007 km$^{-2}$yr$^{-1}$ and 0.096 km$^{-2}$yr$^{-1}$ depending on the regional zone (wet, moderate, dry, and alpine) as identified by Guthrie (2005b). Guthrie (2005b) further observed that more than two-thirds of all landslides below the alpine zone are debris flows.

Cowichan Lake (Figure 16) is an elongated bedrock controlled lake on Southern Vancouver Island. The lake fills the glacially scoured contact between relatively competent Karmutsen and Bonanza volcanic rock on the south shore, and more erodible vol-

5 canic and volcaniclastic rocks of the Sicker group on the north shore (Guthrie, 2005b). The steep northern slopes of Cowichan Lake lie within the dry zone and subsequently the lower range of landslide occurrence (0.004 km$^{-2}$yr$^{-1}$), modified by the underlying bedrock to as much as 0.008 km$^{-2}$yr$^{-1}$ (one landslide 125 km$^{-2}$yr$^{-1}$) (Guthrie, 2005b).

The lowest slopes in the Cowichan Lake valley, adjacent to the shore, are home to approximately 1700 people, and 240 homes were identified as occupying an extreme risk zone related to potential landslide runout (Ebbwater and Palmer, 2019). A

10 landslide runout model was needed to differentiate modern debris flow runout zones from paraglacial fans and the floor of the U-shaped valley, and better discretize risk.

### 3.3.2 Calibrating the Model

Modeled landslides were initiated using LABS in each of four preassigned zones related to likelihood of encountering land-

15 slides anywhere along their path. Those zones represented an encounter probability of 1:100, 1:300, 1:1,000, and 1:3,000 based on air photograph and terrain interpretation (Palmer, 2018).

Landslide initiation locations were created by importing randomly distributed points, a uniform distribution of points, and manually in the GIS tool within LABS by looking at the LiDAR image and choosing initiation zones based on experience in similar areas. The results of each run were compared in a calibration exercise.

The model was calibrated by simulating landslides within the study area, comparing the results to mapped and expected landslide behavior, and checking the modeled landslide magnitude-frequency characteristics against mapped landslide magnitude-frequency characteristics for other areas in coastal British Columbia.

The results showed comparable landslide behavior to mapped landslides (Figure 17) and magnitude frequency curves that are similar to other coastal BC data sets (Figure 18) with a similar rollover and distributions. The modeled results are somewhat

smaller, on average, than those mapped elsewhere in coast, however, that may simply be a result of the limited initiation zone for each modeled landslide (15 m x 15 m), and imposed model restrictions on landslide spreading above 27° slopes. That said, the average modeled landslide size was similar to the average mapped landslide size for this area.

Modeled runout, tended to be a little longer than mapped landslide runout, however, that could be a result of more landslides initiated at one time, the cumulative footprint of multiple runs, or the improved ability to follow a modeled landslide in a

30 channel compared to a similar sized mapped event.



### 3.3.3   Calculating Landslide Runout

Once tested, 1,364 new landslide initiation points were selected across the 1:100, 1:300, 1:1,000, and 1:3,000 encounter probability zones (Figure 19). A user-based initiation-point selection method was used for the final model runs as this method generally resulted in landslide generation somewhat more frequently than randomly or uniformly generated points that would

5   sometimes occur on a flatter portion of the slope. LABS then automatically created initiation zones (as a selected option) of nine agents in 15 m by 15 m grids (where grid cells are 5 m a side -see Figure 20).

Fifty landslide runs were modeled from each landslide initiation zone (both random and manually selected). Though view-

10   able in LABS, the results were exported as Geotiffs to enable visualization in Google Earth and ArcGIS software (Figure 21). Over 70,000 debris flows were modeled and a distinct runout limit was derived.

The result of the landslide runs was, with few exceptions, that the cumulative footprint of modeled landslides did not reach residential homes on the paraglacial fans. Instead, landslides tended to terminate on upper- and mid-fan slopes that were between 10 and 20 degrees (Figure 22).

The likelihood of any individual landslide reaching the runout limit can be explored in LABS by obtaining information about the number of times any pixel was inundated out of the total number of runs. In this instance, however, the total runout limit was more practical (Figure 23).

### 3.3.4   Estimating Likelihood of Damage

The probability of damage due to debris flows and debris avalanches has been discussed by several authors and can be modeled empirically (Jakob et al., 2012; Papathoma-Kohle et al., 2012), analytically (Corominas et al., 2014; Mavrouli et al., 2014), or using engineering judgment (Winter et al., 2014). Ciurean et al. (2017) developed an analytical method that required only

25   depth, and compared favorably to both empirical and analytical methods previously developed.

The latter method was used to estimate the potential impact of debris flows or debris avalanches that were modeled to reach buildings. A damage class was assigned to each polygon based on estimated landslide depth from the LABS model (Table 5) and potential degree of loss was determined from shown in Figure 4-6 for different classes ofbuildings.



### 3.3.5   Case II Summary

LABS was used to model debris flow runout from steep slopes above a community on the north shore of Cowichan Lake.

With few exceptions, the cumulative footprint of modeled landslides did not reach residential homes on the paraglacial fans.

Exceptions were easily identified on two types of maps, a runout limit map and a potential damage map that relates to building vulnerability.

With over 70,000 landslide runs, the probability that a modeled debris flow will exceed the distal limit indicated on the maps
was less than 0.000015.

Properties above (north) of the distal limit of modeled runout can use the potential damage curves to inform subsequent investigation.

### 4 Discussion and Limitations of Use

*"All models are wrong, but some are useful."* — George E.P. Box

Both case studies demonstrate the potential usefulness of an easily employed, regional runout model. LABS is predictive and, at least for shallow, rapid to extremely rapid, flow-type landslides, appears to provide viable runout results, as well as information about landslide depth, area (footprint) and volume.

However, LABS is still limited to the rules for erosion and deposition employed, and experiments in other regions of the world will benefit users.

Some of the potential pitfalls of the program are articulated below.

### 4.1   Depth variability

As an artifact of the rules, individual runs may exhibit sudden deposition or scour along their path in excess of what would actually be expected. This occurs when a single agent at a pixel picks a low probability depth for either scour or deposition. Multiple model runs are therefore recommended and should provide better depth results because individual highs and lows are
averaged out.

### 4.2   Parameter sensitivity

There are considerable opportunities to tweak landslide behavior within the program. The authors recommend the settings shown in Figure 12 and Figure 20 as default values. The *Fan Maximum Slope* slider is intuitive, as is the *Mass Loss* slider,





however, if adjusting the other sliders, results should be checked using the inspector tool, for depths as well as the overall footprint.

## 4.3   Linearity

Very steep slopes may produce a strong linear landslide orientation, easily seen when multiple landslides are triggered. This
occurs when the DEM at the model resolution (5 m) is so steep that it overwhelms the path selection at each time step and spreading has not yet occurred (recall that spreading is has a user defined slope limit). While natural analogs are readily found (e.g. Figure 26), the modeled results may nonetheless bypass local topographic effects and choose paths that vary somewhat from the real-world equivalent. DEM effects have been noticed by others; Degetto et al. (2015) and Stolz and Huggel (2008) both demonstrate that the DEM source can dramatically influence the outcome of debris flow models, even  at
equivalent resolutions. Horton et al. (2013) propose that a 10 m resolution DEM is appropriate for regional mapping. In our case, we suggest that the 5 DEM strikes the right balance between processing power and reasonable results, and LABS has been optimized such that the agents work on a 5 m cell size.

## 4.4   Debris flows vs debris floods

Despite considerable literature, confusion about the difference between debris flows, debris floods, and hyperconcentrated flows persists (Pierson, 2005; Calhoun and Clague, 2018; Keaton, 2019). Geomorphic criteria for distinguishing between debris flows and debris floods such as those derived by Wilford et al. (2004) may not fully align with other defining criteria such as sediment concentration and shear strength (Figure 27).

LABS simulates rapid to extremely rapid landslides of the flow type, but is not intended to model debris floods or hyper-
concentrated flows. LABS may, therefore, underestimate runout of channelized debris flows particularly those channels that are transitional to debris floods. However, as demonstrated in our case study, LABS can provide volumetric sediment supply to channels that can be subsequently modeled using the right tool. Further, within its intended parameters (the empirical observations of scour and deposition) LABS tends to show the depositional extent of debris flows in channels that might be otherwise lost to other processes. Nonetheless, the process difference should be recognized by the reader.

## 4.5   Detailed simulation

LABS is a regional tool based primarily on empirical observations of aggregate debris flow behavior, particularly scour and deposition along the landslide path. Its probabilistic nature will result in similar but different outputs from one run to the next. We would expect nature to behave much the same way. However, if a detailed analysis of a single debris flow is sought, this tool





is neither at the correct scale, nor does it adequately address the site specific controls such as rheology, detailed topography, moisture content, and geology (among other factors). Indeed, LABS explicitly seeks to ignore these factors in order to provide a practical regional tool. For detailed analysis, the reader is directed to any of several excellent dynamic models.

## 5    Conclusions

In order to address a perceived need for a debris flow or debris avalanche runout model that can be applied regionally with relatively few inputs, we developed, and present herein, an agent-based landslide-simulation model called LABS.

LABS is a fully predictive model whereby autonomous sub-routines, or agents, act on an underlying DEM using a set of probabilistic rules for scour, deposition, path selection, and spreading behavior. Along the way, agents keep track of the changes they make to the DEM, of the landslide to which they belong, of nearby (adjacent) agents, and of their own mass balance.

We demonstrate the use of LABS in two case studies.

In the first, we used LABS to determine the sediment input (in $m^3$) to a stream network in the steep mountains of Indonesia's province of Papua. Sediment input was used to bulk the hydrographs for subsequent debris flood modeling (not shown) at specified return periods.

In the second case study, we used LABS to predict runout distance in a residential community on Vancouver Island, Canada.

By running tens of thousands of landslides, we defined a modeled landslide runout limit and demonstrated that most houses were beyond the threat of debris flow runout. For those that remained in the runout zone, we used the average depth information to assign potential damage curves to unprotected properties.

LABS is freely available to non-profit groups (universities, NGOs, government) and may be downloaded here **[ADDRESS TO BE PROVIDED IN FINAL SUBMISSION]**.

LABS simplifies extremely complex behavior to provide reasonable predictions of outcomes. Should there be a perceived difference between modeled results, and on-the-ground evidence, the ground-based evidence should take priority. LABS does not relieve professionals from using their experience, training and education to make good judgments when assessing actual ground conditions, but provides additional understanding of processes and credible outcomes.



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



**List of Figures**







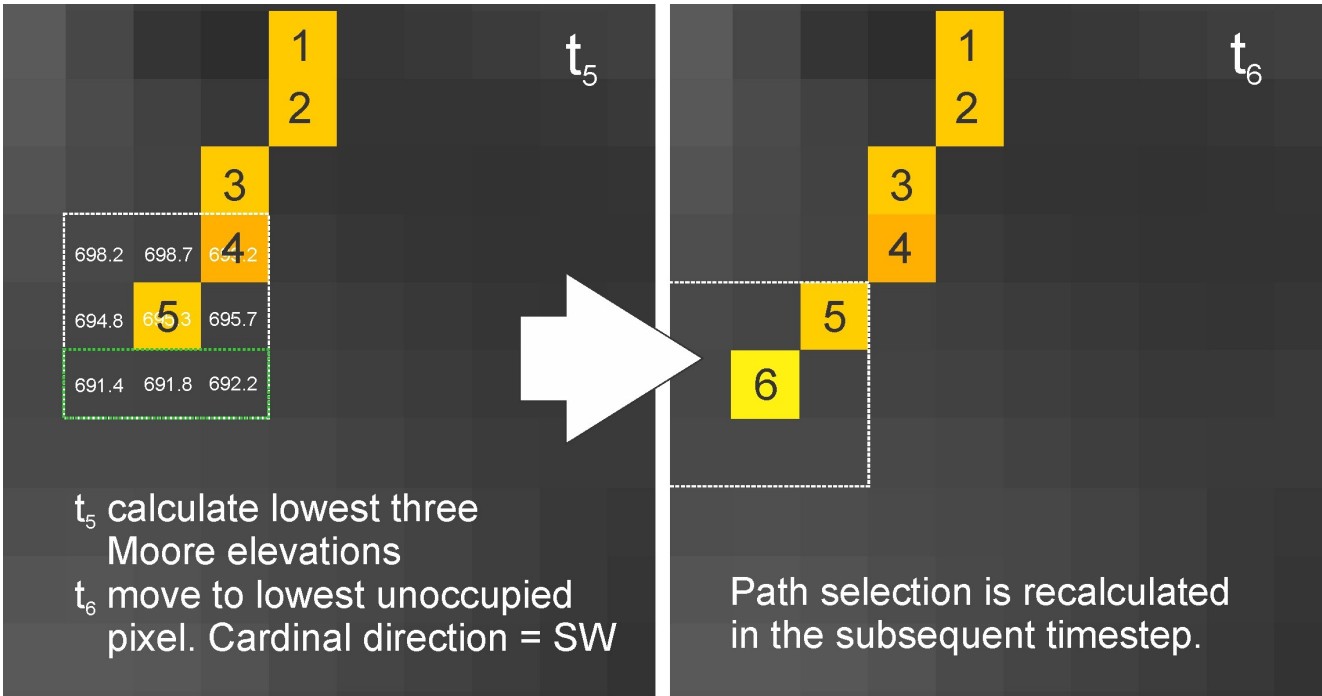

**Figure 1. Path selection based on aspect determined by Moore neighbors. Black numbers represent time steps, and white numbers represent actual elevations. Example is from Lake Cowichan on Vancouver Island. Pixel resolution is 5 m.**

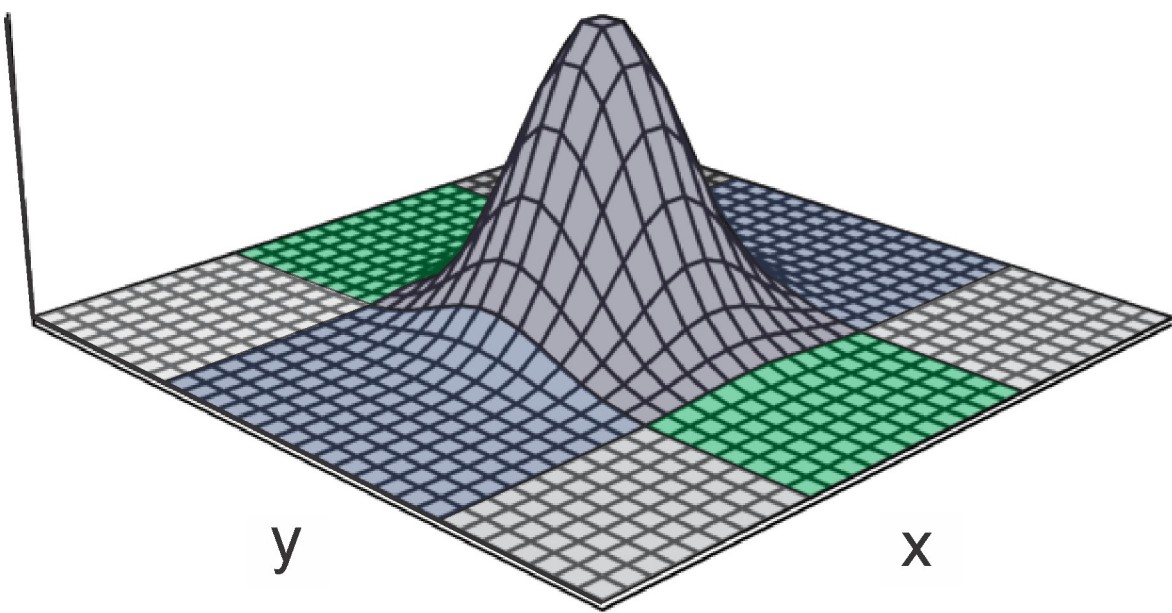

**Figure 2. An example surface plot of a normal probability density function where mass from the center pixel (each pixel is uniformly colored) might be redistributed (spawning new agents) amongst adjacent pixels.**





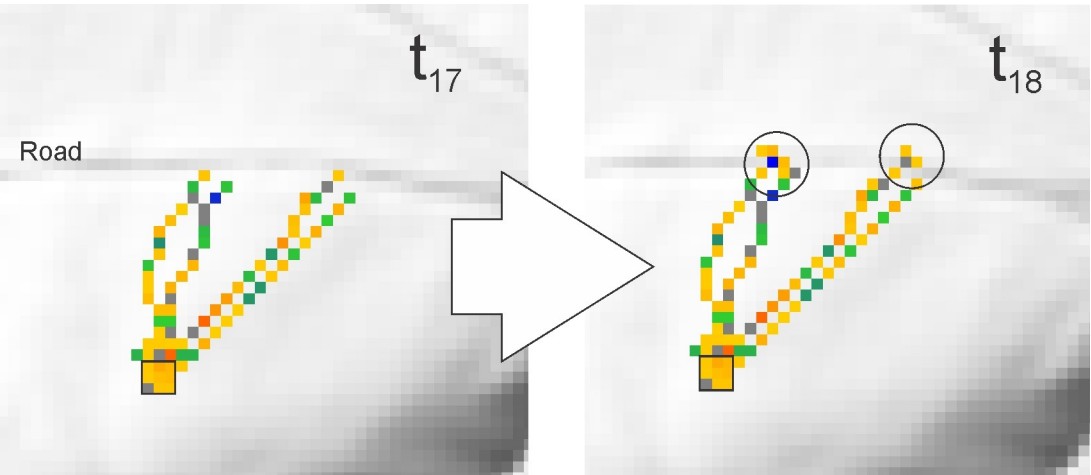

**Figure 3. Agent spawning (inside circles) due to a topographic change at a road that is reached in time step 18. Landslide is traveling NE (to upper right corner) and the initiation zone is the 15 x 15 m square outlined in black.**

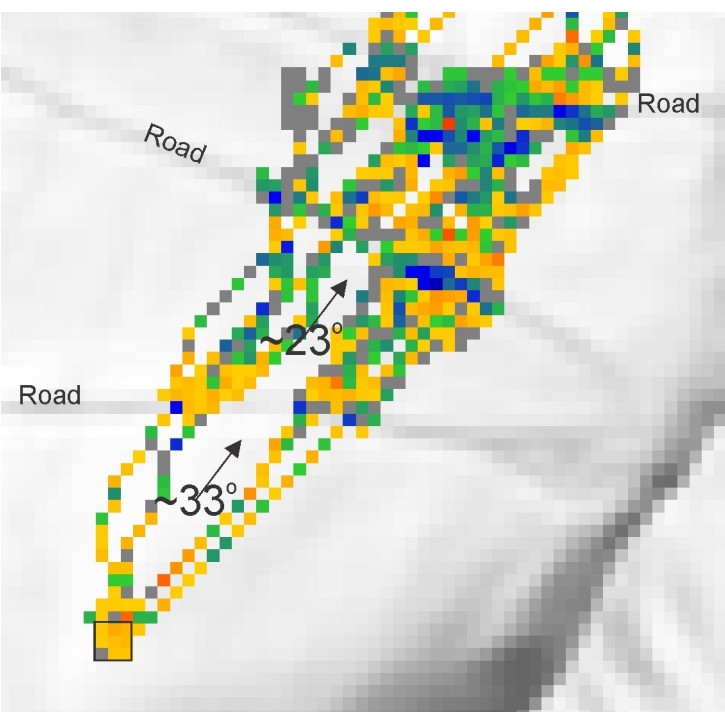

**Figure 4. Topographic effects on landslide propagation is easily seen. Agents spawn at and below the road(s), causing the landslide to spread. Approximate average slopes are shown on the figure. This figure is the continuation of the landslide path fromFigure 3. The landslide is moving NE (towards the top right) and the initiation zone is the 15 x 15 m square outlined in black in the lower left corner.**



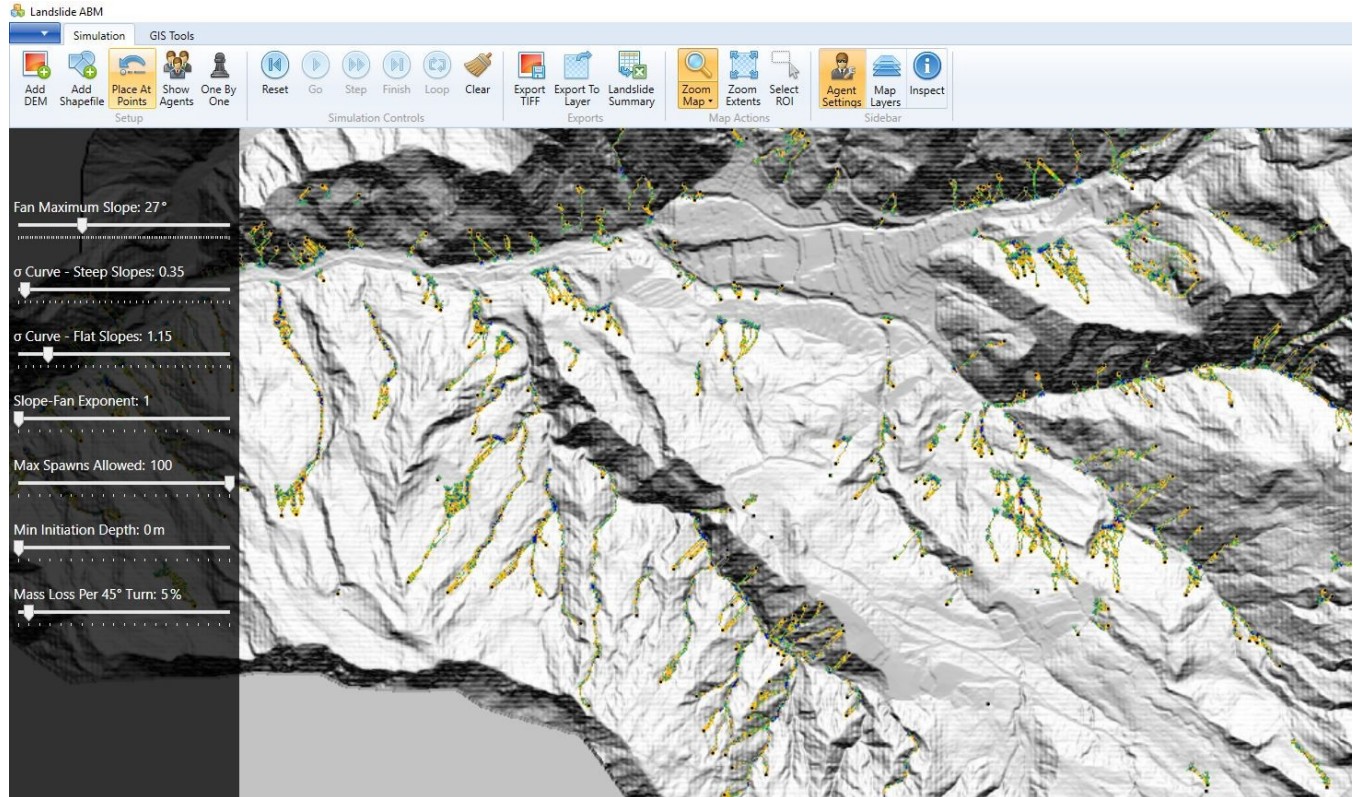

**Figure 5. Example output within LABS for a single run, multiple landslides. Example site is in Indonesia.**

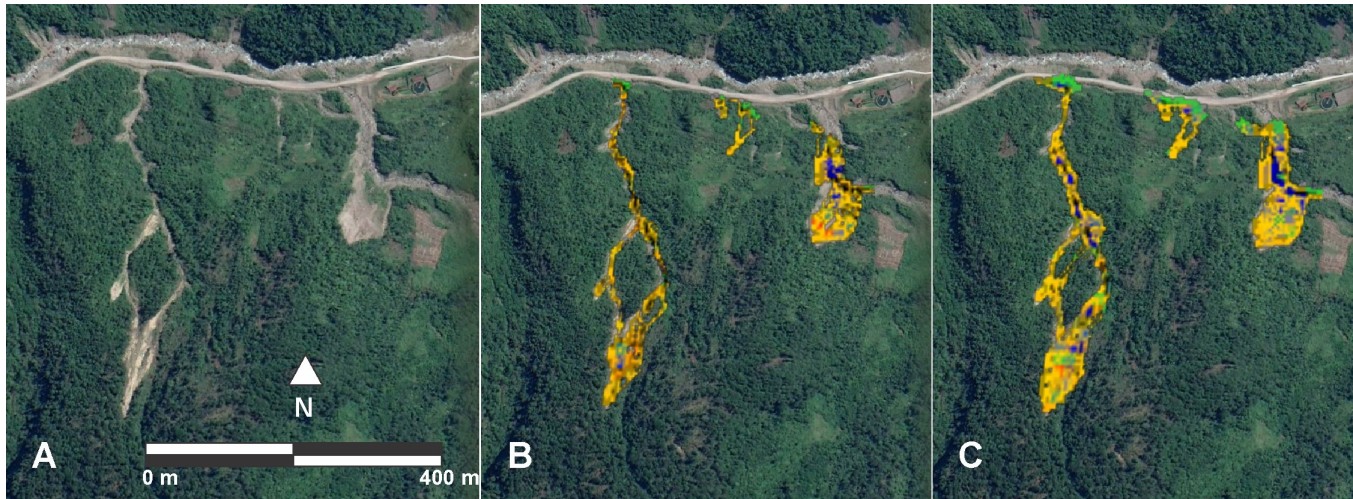

**Figure 6. Comparison between historic landslides (A), modeled landslides from a single run (B), and (C) from 50 runs (the cumulative footprint). Background image © Google Earth. Example from a site in Indonesia.**





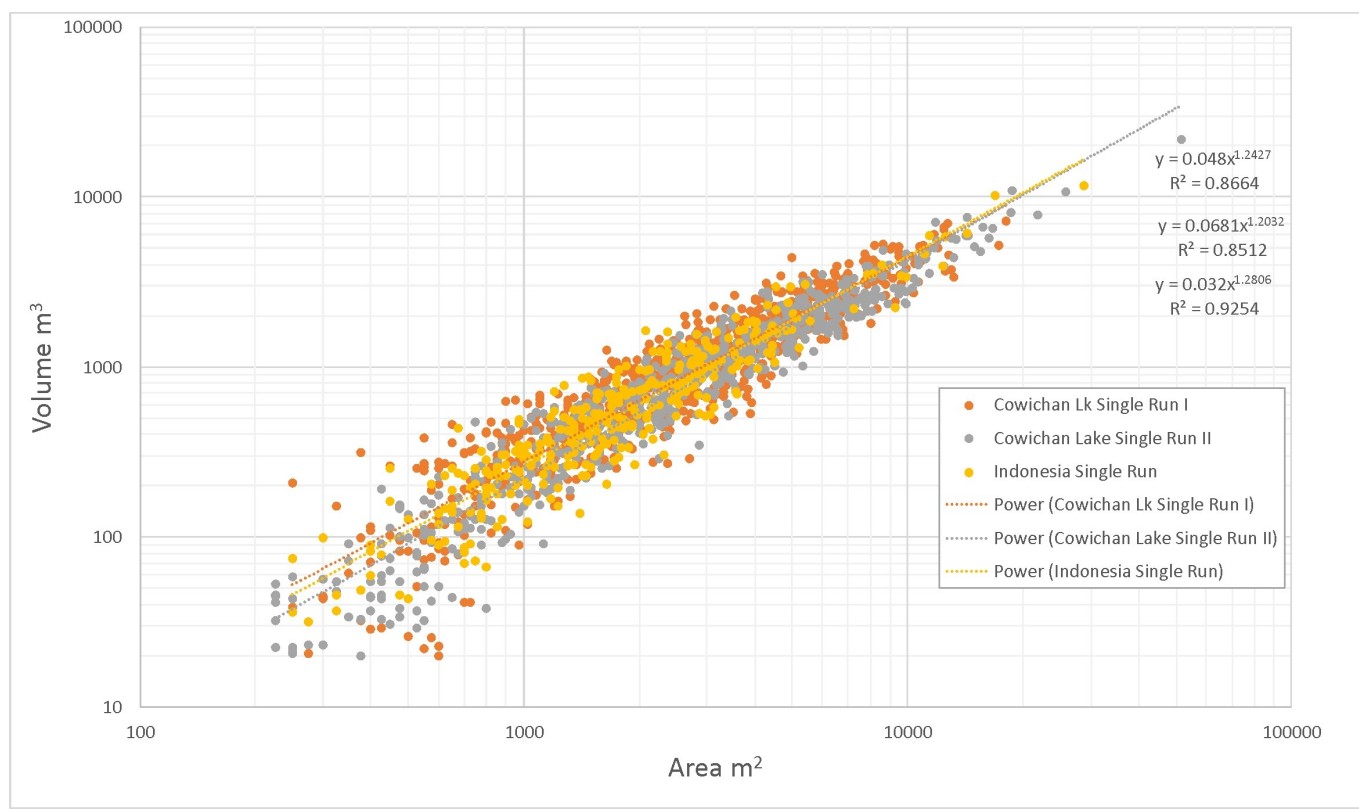

**Figure 7. Landslide area volume relationships for modeled single run landslides in Indonesia and from Vancouver Island. Data come from case studies described in subsequent sections.**




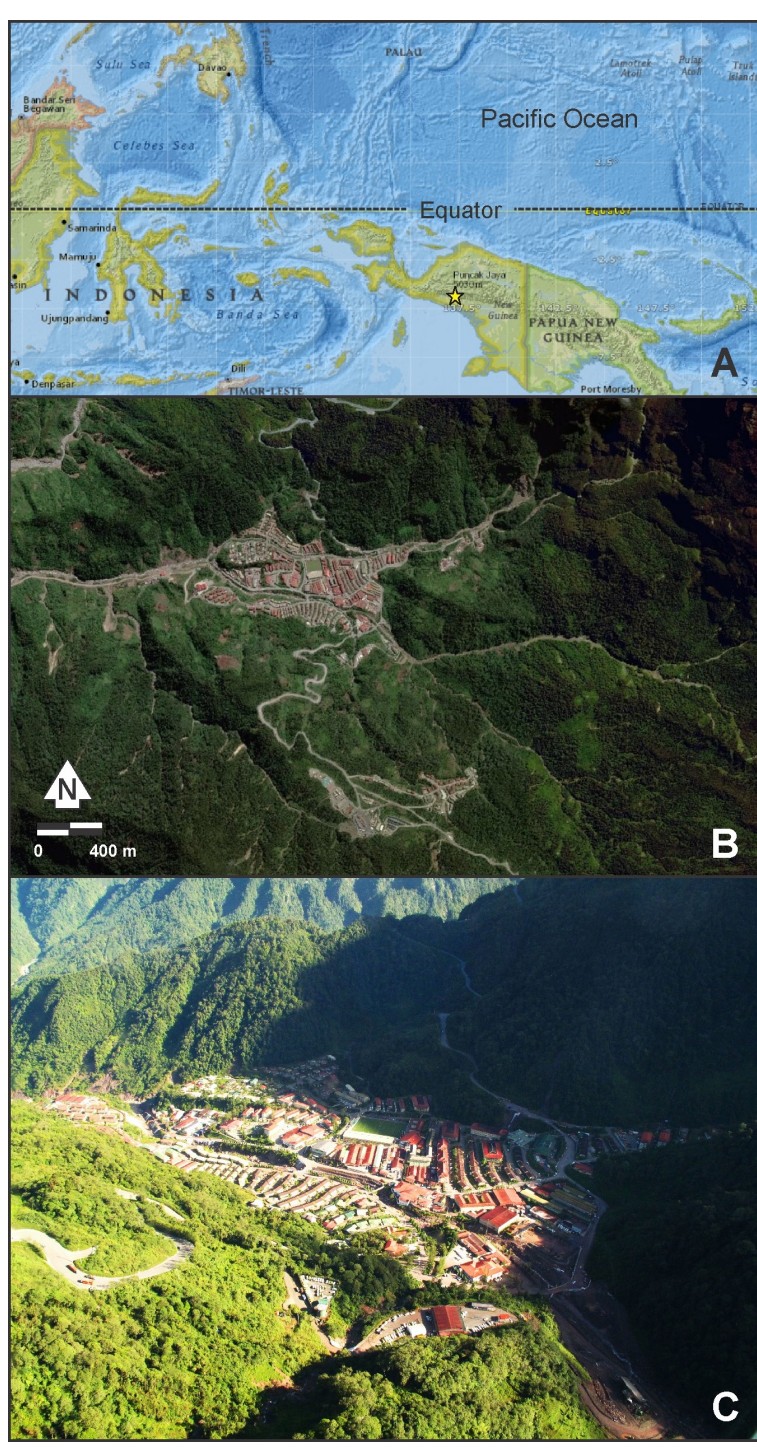

**Figure 8. Tembagapura located in the province of Papua, Indonesia. (A) Regional view, yellow star indicates Tembagapura (Image © ESRI and National Geographic World Map), (B) vertical image over Tembagapura showing debris flows and debris floods (Image © Google Earth), (C) Oblique view of Tembagapura in the steep Jayawijaya Mountains.**



**Figure 9. An historical inventory revealed 375 landslides, primarily debris flows, in the slopes surrounding Tembagapura. Years refer to the year of imagery and the background image is a veritical air photograph obtained by the authors.**

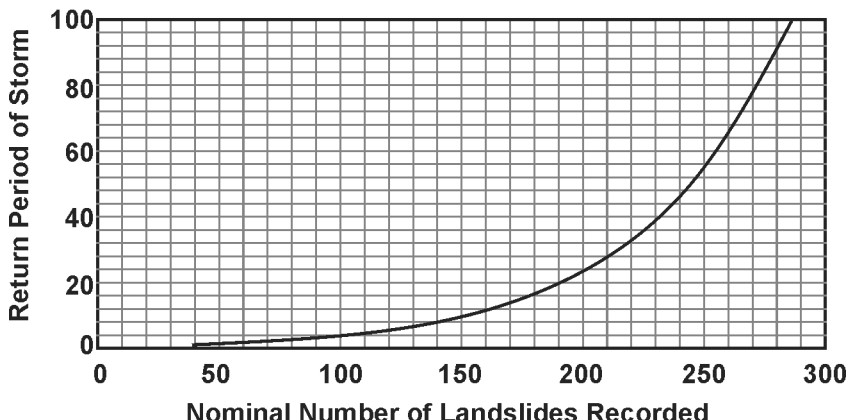

**Figure 10. The relationship between landslides occurrence and landslide generating rainfall at Tembagapura.**





**Figure 11. Landslide initiation zones painted at the estimated source of mapped debris flows (darker line work) in LABS.**







**Figure 12. A single run of the landslides whose initiation zones were painted in Figure 11 above. Colors relate to scour (yellow to red where red is deeper) and deposition (green to blue where blue is deeper)**



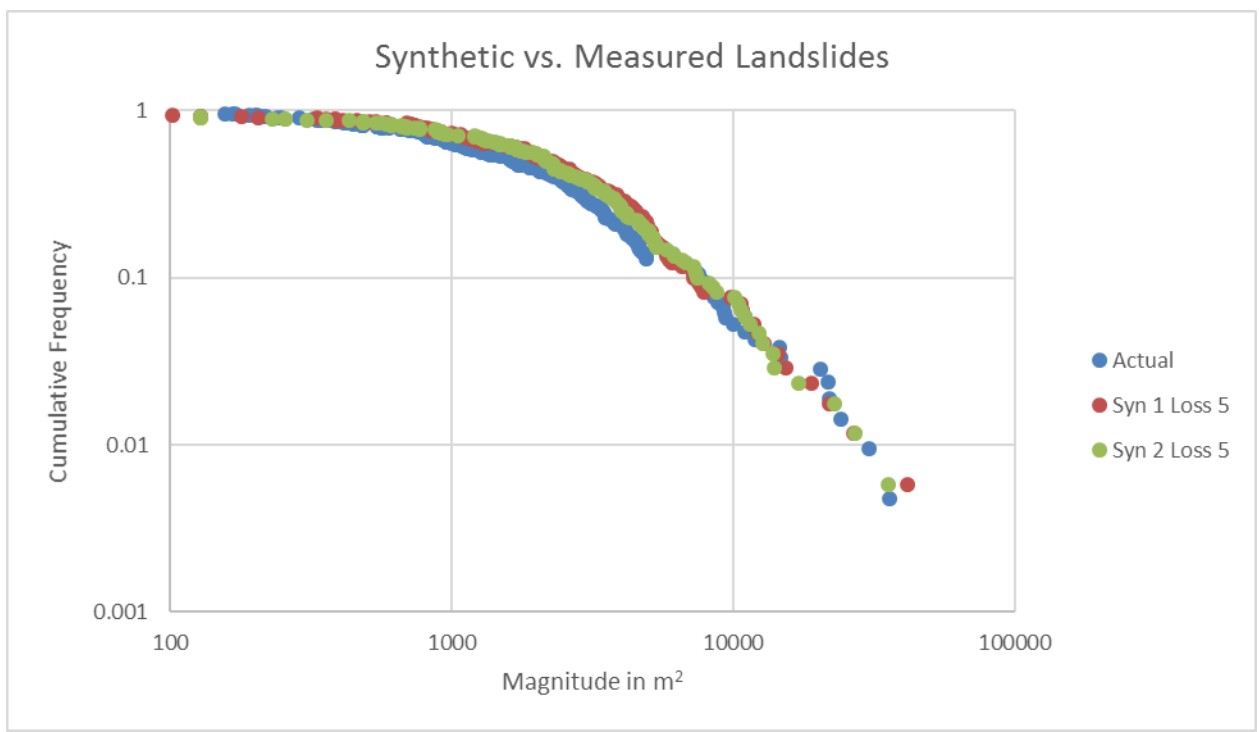

**Figure 13. Cumulative magnitude frequency comparison between mapped and modeled landslides (two runs).**



Figure 14. Landslides simulated from random points, stratified by susceptibility, above the slopes of Tembagapura.



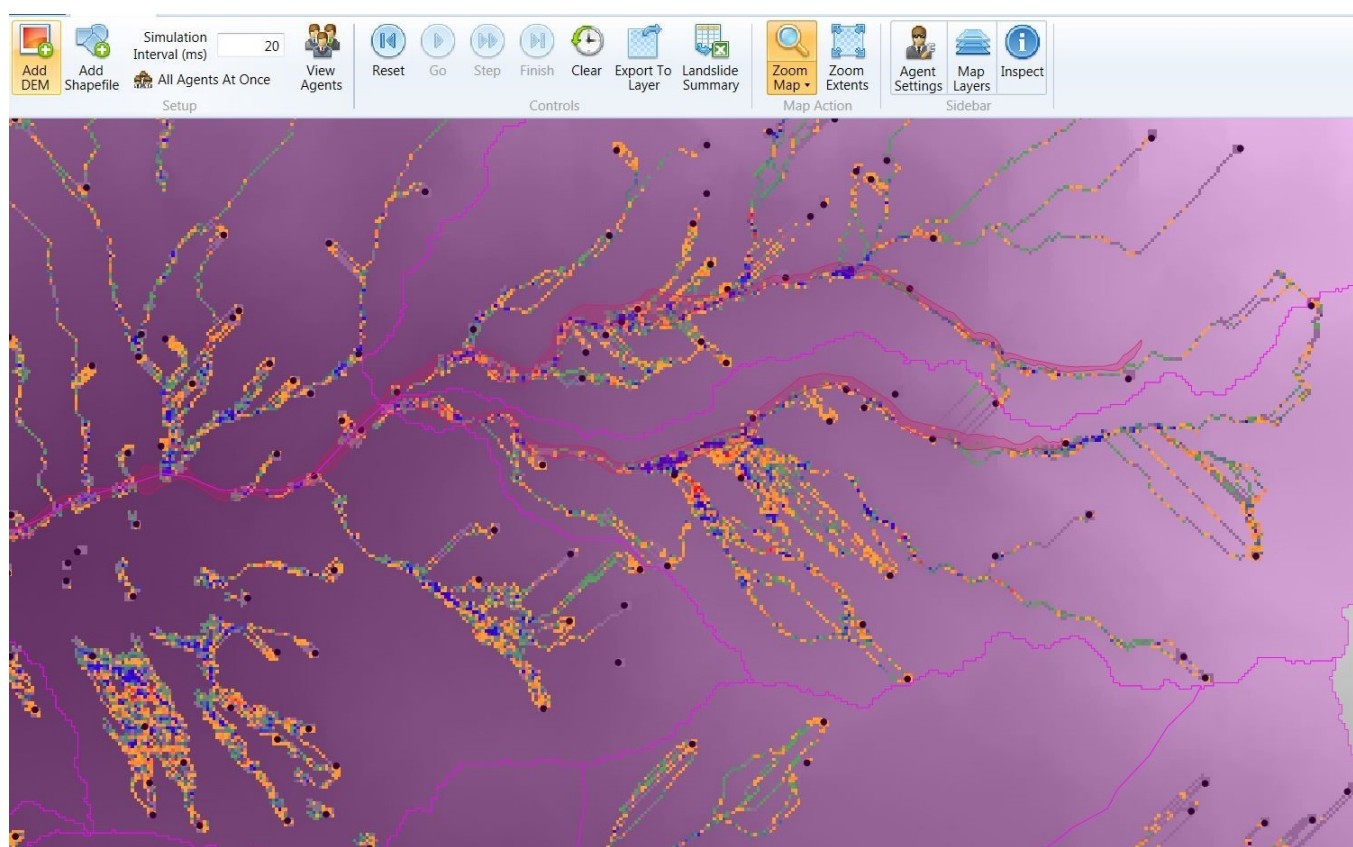

Figure 15. Detail of the simulation shown in Figure 14.

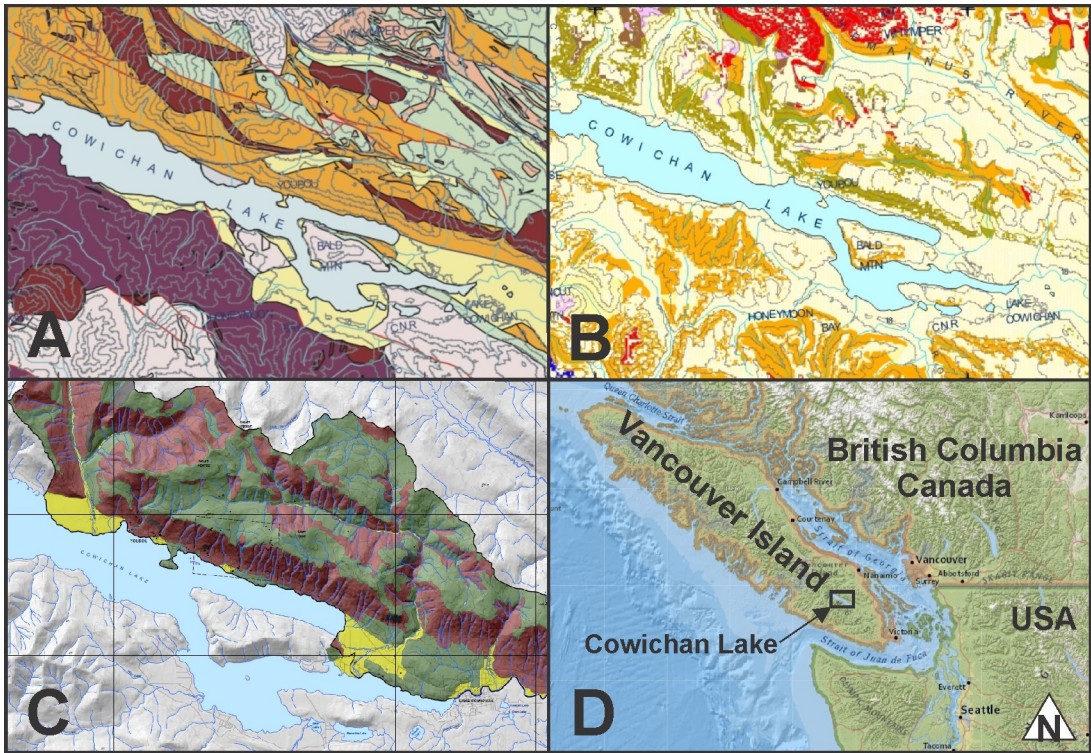

**Figure 16. Cowichan Lake on Vancouver Island showing (A) Bedrock Geology and volcanics of the Sicker Group (orange), Bonanza Group (purple) and Karmutsen Formation (pink); (B) Mass movement potential (0.004 landslides $km^{-2}yr^{-1}$ for the orange zone, up to double that for the tan zone on the north side of the lake); (C) Surficial geology (colluvium, till, and fluvial gravels mapped as purple, green and yellow respectively); and (D) the overall location. Figure taken from Guthrie (2005a) (A) and (B) and Palmer (2018) (C).**



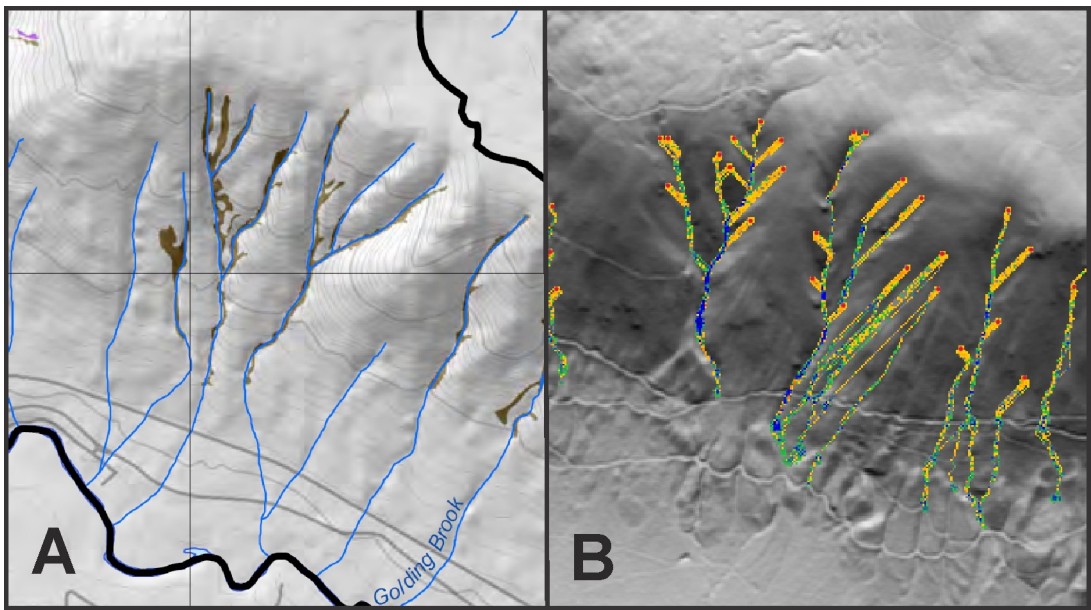

**Figure 17. A comparison of mapped debris flow paths (A) and modeled debris flow paths (B) in the study area.**





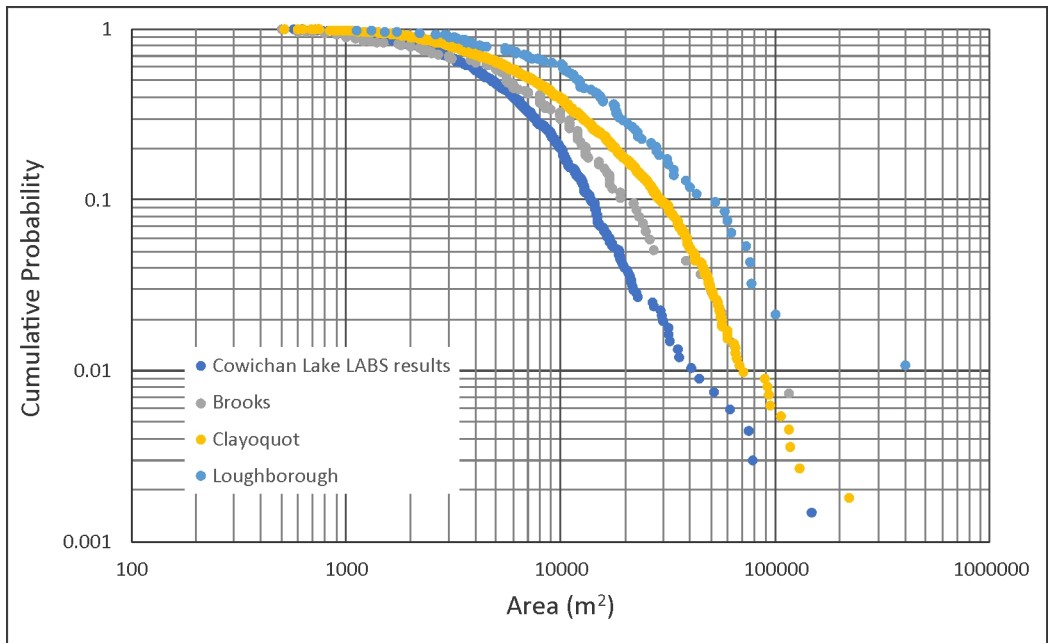

**Figure 18. Cumulative magnitude frequency curves from Cowichan Lake calibration results, and mapped landslides elsewhere in coastal BC. Data from Guthrie and Evans (2004a, b).**





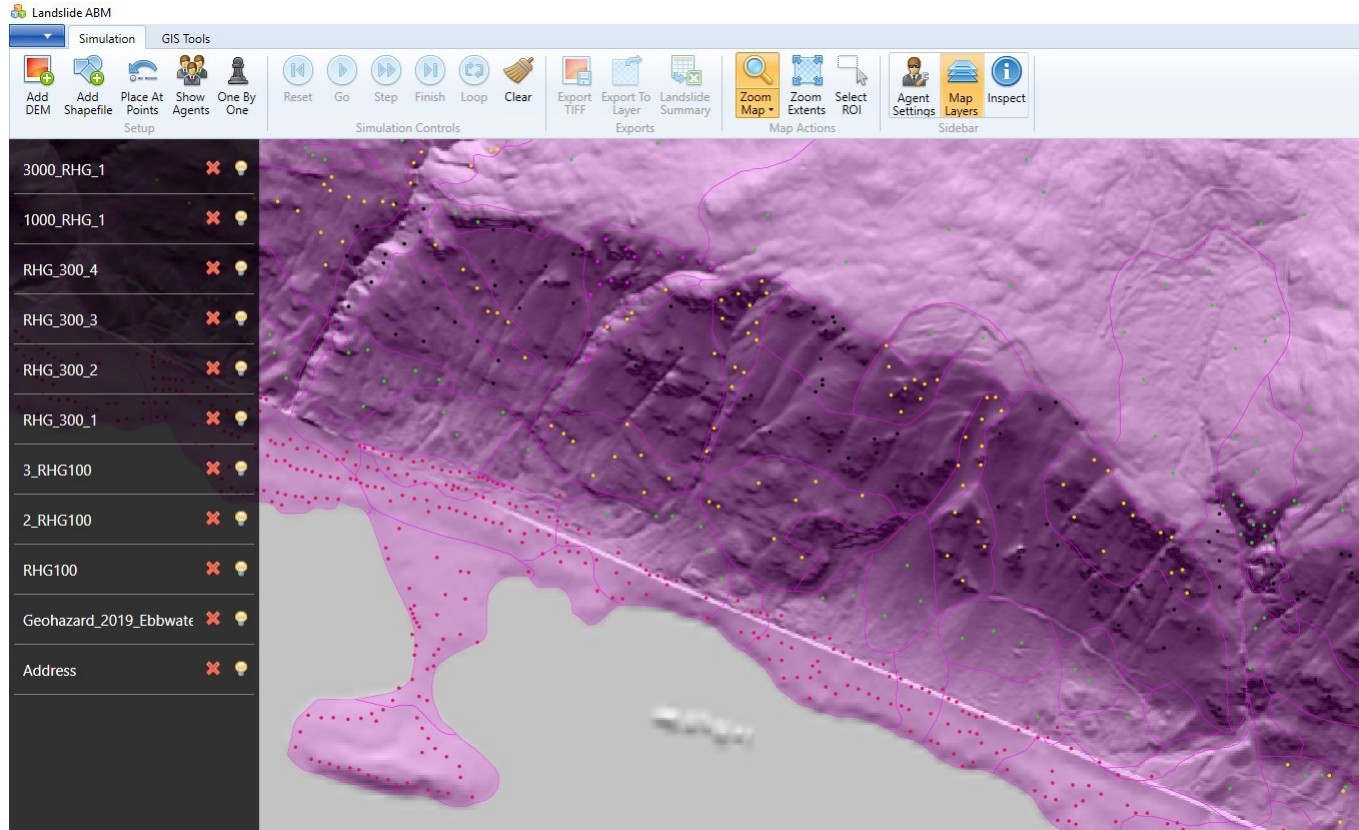

**Figure 19. Landslide initiation point locations for different encounter probability polygons (polygons are the pink translucent overlay). Residential addresses are denoted by the pink dots.**



**Figure 20. A close up view of landslide initiation points and the LABS generated 15 m x 15 m initiation zones. Each zone contains nine agents. The sliders on the left control agent behavior as explained in the Methods section.**

**Figure 21. Modeled landslides along the north shore of Cowichan Lake from different encounter probability polygons, and overall. Results can be imported into Google Earth, as shown here, for convenient visualization (background image is © Google Earth).**

**Figure 22. Oblique Google Earth view of LABS results showing the runout limit of thousands of modeled debris flows (background image is © Google Earth).**



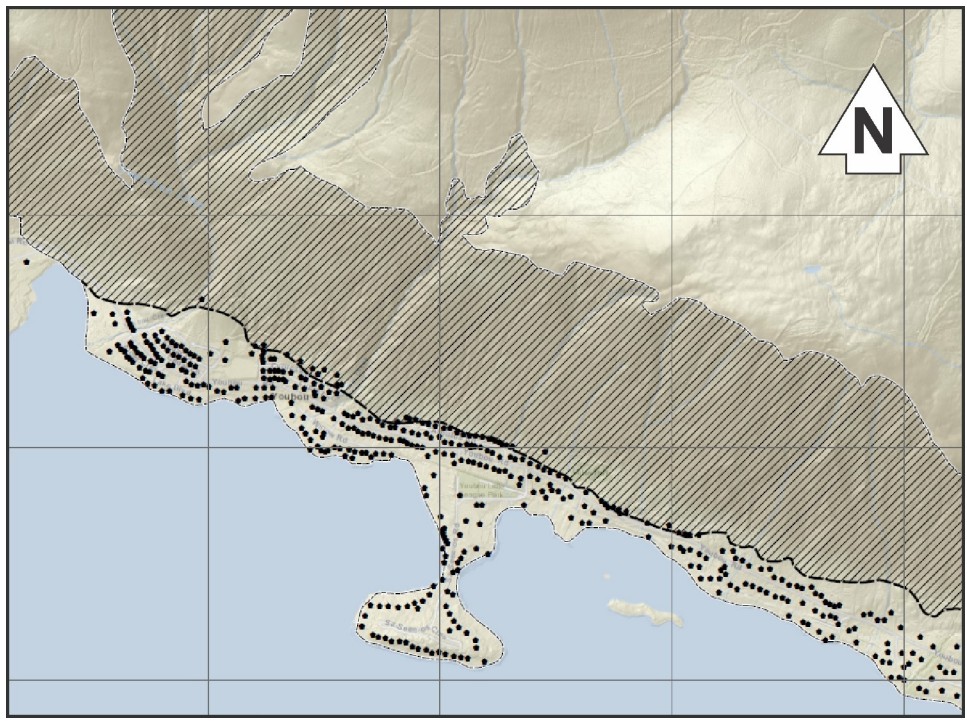

**Figure 23. Modeled debris flow runout limit. Note that most of the 240 properties (black pentagons) are outside the modeled runout zone. Background image © ESRI World Topographic Map.**


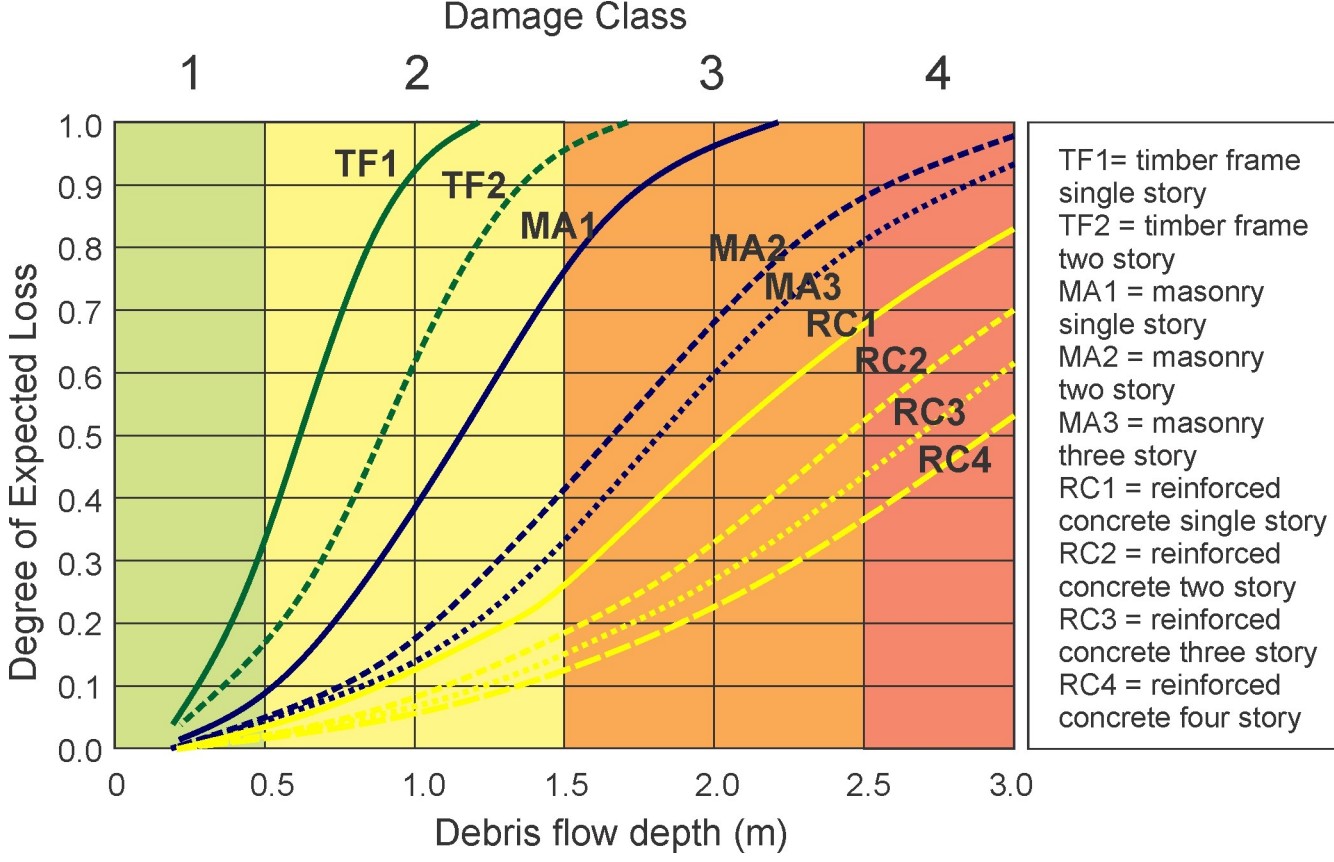

**Figure 24. Vulnerability identified by degree of expected loss for constructed buildings by debris flow depth (Ciurean et al., 2017).**



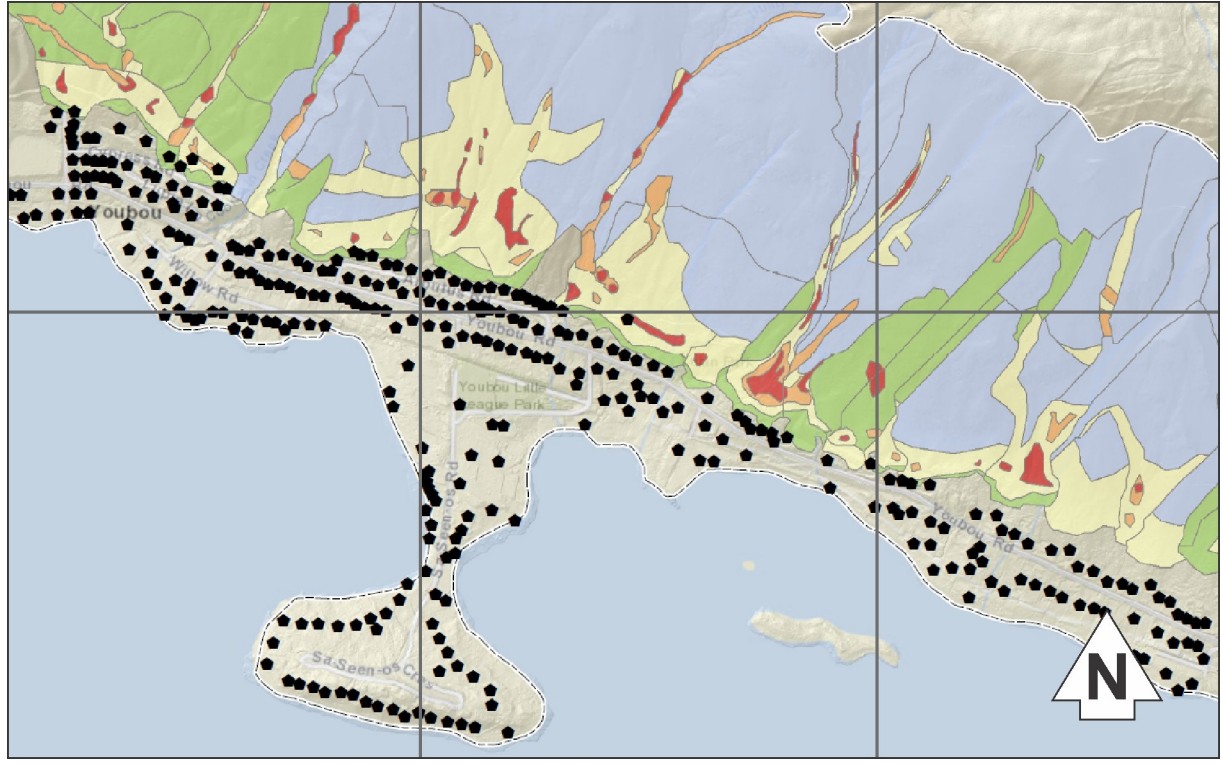

**Figure 25. Damage map for the north shore of Cowichan Lake. The colors (other than blue) match Figure 24. Blue represents scour and transport zones and construction is deemed unlikely. Background image © ESRI World Topographic Map.**

**Figure 26. Strong linear orientation of landslide tracks on steep slopes in Indonesia (A) where multiple landslides occur at once, and on the North Shore of Cowichan Lake (B) where multiple landslides were modeled to occur at once (background images © Google Earth).**


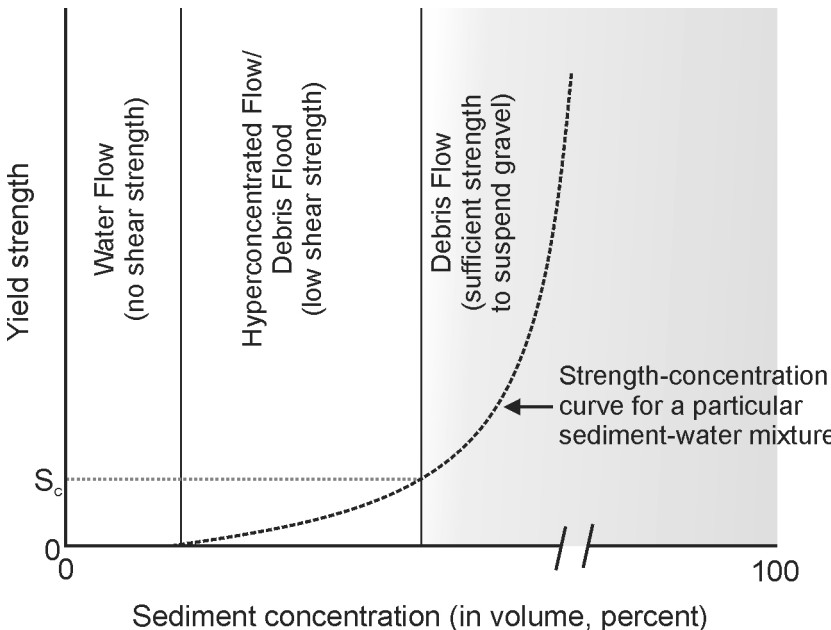

**Figure 27. Debris flows as expected to be modeled by LABS (shaded area). $S_c$ represents the critical shear strength beyond which gravel (4 mm or larger) is suspended. Figure modified slightly from Pierson (2005).**



**List of Tables**





**Table 1. Basic scour and deposition rules used in LABS. Data comes from Wise (1997); Guthrie et al. (2008, 2010)**

Probability (P) of scour or deposition by slope bins

| Scour depth (m) | P | Deposition (m) | P |
|---|---|---|---|
| 0° − < 10° | | | |
| 0 | 0.96 | NA | NA |
| 0.2 | 1 | 0.2 | 0.04 |
| NA | NA | 0.96 | 0.4 |
| NA | NA | 1.38 | 0.24 |
| NA | NA | 2.06 | 0.32 |
| 10° − < 16° | | | |
| 0 | 0.82 | NA | NA |
| 0.31 | 0.1 | 0.41 | 0.12 |
| 0.82 | 0.081 | 0.95 | 0.33 |
| NA | NA | 1.46 | 0.28 |
| NA | NA | 2.26 | 0.27 |
| 16° − < 21° | | | |
| 0 | 0.37 | 0 | 0.25 |
| 0.39 | 0.46 | 0.46 | 0.22 |
| 0.9 | 0.16 | 0.94 | 0.28 |
| 1.4 | 0.01 | 1.37 | 0.08 |
| NA | NA | 2.08 | 0.17 |
| 21° − < 27° | | | |
| 0 | 0.15 | 0 | 0.46 |
| 0.37 | 0.48 | 0.36 | 0.3 |
| 0.9 | 0.3 | 0.89 | 0.14 |
| 1.43 | 0.05 | 1.41 | 0.06 |
| 2 | 0.02 | 2 | 0.04 |
| 27° − < 33° | | | |
| 0 | 0.14 | 0 | 0.62 |
| 0.38 | 0.42 | 0.31 | 0.21 |
| 0.88 | 0.29 | 1 | 0.12 |
| 1.36 | 0.08 | 1.4 | 0.03 |
| 1.97 | 0.07 | 2 | 0.02 |
| 33° − < 39° | | | |
| 0 | 0.04 | 0 | 0.88 |
| 0.37 | 0.49 | 0.37 | 0.08 |
| 0.94 | 0.32 | 0.8 | 0.04 |
| 1.31 | 0.14 | NA | NA |
| 2 | 0.01 | NA | NA |
| 39° − < 46° | | | |
| 0 | 0.3 | 0 | 1 |
| 0.35 | 0.6 | NA | NA |
| 0.95 | 0.05 | NA | NA |
| 1.5 | 0.02 | NA | NA |
| 1.99 | 0.03 | NA | NA |
| 46° − < 60° | | | |



| | | | |
|---|---|---|---|
| **Table 1 continued** | | | |
| Probability (P) of scour or deposition by slope bins | | | |
| 0 | 0.65 | 0 | 1 |
| 0.1 | 0.34 | NA | NA |
| 0.35 | 0.01 | NA | NA |
| 60°+ | | | |
| 0 | 0.96 | 0 | 1 |
| 0.1 | 0.04 | NA | NA |
| −10°− < 0° (opposing slope) | | | |
| 0 | 1 | NA | NA |
| NA | NA | 0.2 | 0.04 |
| NA | NA | 0.96 | 0.4 |
| NA | NA | 1.38 | 0.24 |
| NA | NA | 2.06 | 0.32 |
| −33°− < −10° (opposing slope) | | | |
| 0 | 1 | NA | NA |
| NA | NA | 0.96 | 0.04 |
| NA | NA | 1.38 | 0.4 |
| NA | NA | 2.06 | 0.24 |
| NA | NA | 3 | 0.32 |
| < −33° (opposing slope) | | | |
| 0 | 1 | NA | NA |
| NA | NA | 1.38 | 0.04 |
| NA | NA | 2.06 | 0.4 |
| NA | NA | 3 | 0.24 |
| NA | NA | 5 | 0.32 |



**Table 2. Areas and volumes from this study compared with historical studies of debris flow area volume relationships**

| Equation | Min Area (m²) | Max Area (m²) | $n$ | Source |
|---|---|---|---|---|
| Debris flows | | | | |
| $V=0.596A^{1.02}$ | $0.6 \times 10^1$ | $2.1 \times 10^3$ | 930 | Cha et al. (2018) |
| $V=0.155A^{1.09}$ | $7 \times 10^2$ | $1.2 \times 10^5$ | 124 | Guthrie and Evans (2004b) |
| $V=0.19A^{1.19}$ | $5 \times 10^1$ | $4 \times 10^3$ | 11 | Imaizumi et al. (2008) |
| $V=0.39A^{1.31}$ | $1 \times 10^1$ | $3 \times 10^3$ | 51 | Imaizumi and Sidle (2007) |
| $V=1.036A^{0.88}$ | $2 \times 10^2$ | $5.2 \times 10^4$ | 615 | Martin et al. (2002) |
| $V=0.048A^{1.24}$ | $2.5 \times 10^2$ | $2.9 \times 10^5$ | 353 | Modeled, this study: Indonesia manually selected initiation zones |
| $V=0.0681A^{1.20}$ | $2.5 \times 10^2$ | $1.8 \times 10^5$ | 797 | Modeled, this study: Vancouver Island manually selected initiation zones |
| $V=0.032A^{1.28}$ | $2.2 \times 10^2$ | $5.1 \times 10^5$ | 703 | Modeled, this study: Vancouver Island randomly selected initation zones |



**Table 3. Accumulated landslide generated sediment (in m$^3$) between known debris flood years**

| Debris Flood Year | 1998 | 2010 | 2013 | 2014 | 2016 | 2017 |
|---|---|---|---|---|---|---|
| Sub-basin | | | | | | |
| 0 | 16,198 | 24,715 | 5,865 | 2,513 | 5,027 | 5,585 |
| 1 | 20,853 | 31,819 | 7,550 | 3,236 | 6,472 | 7,191 |
| 2 | 14,130 | 21,561 | 5,116 | 2,193 | 4,385 | 4,872 |
| 3 | 53,291 | 81,315 | 19,295 | 8,269 | 16,539 | 18,376 |
| 4 | 17,712 | 27,027 | 6,413 | 2,748 | 5,497 | 6,108 |
| 5 | 14,551 | 22,203 | 5,268 | 2,258 | 4,516 | 5,018 |
| 6 | 23,576 | 35,974 | 8,536 | 3,658 | 7,317 | 8,130 |
| 7 | 2,414 | 3,683 | 874 | 375 | 749 | 832 |
| 8 | 3,294 | 5,026 | 1,193 | 511 | 1,022 | 1,136 |
| 9 | 27,510 | 41,976 | 9,960 | 4,269 | 8,538 | 9,486 |
| 10 | 36,880 | 56,274 | 13,353 | 5,723 | 11,446 | 12,717 |
| 11 | 10,776 | 16,442 | 3,902 | 1,672 | 3,344 | 3,716 |
| Total Watershed | 241,185 | 368,015 | 87,325 | 37425 | 74,852 | 83,167 |




**Table 4. Accumulated landslide generated sediment (in m³) by return period of landslide generating storms.**

| Storm Return Period (years) | 100 | 50 | 25 | 20 | 10 | 5 |
|---|---|---|---|---|---|---|
| Sub-basin | | | | | | |
| 0 | 17,772 | 16,587 | 15,402 | 15,064 | 13,879 | 12,863 |
| 1 | 22,880 | 21,354 | 19,829 | 19,393 | 17,868 | 16,560 |
| 2 | 15,503 | 14,470 | 13,436 | 13,141 | 12,107 | 11,221 |
| 3 | 88,653 | 82,743 | 76,833 | 75,144 | 69,234 | 64,168 |
| 4 | 19,434 | 18,138 | 16,843 | 16,472 | 15,,177 | 14,066 |
| 5 | 15,965 | 14,901 | 13,836 | 13,532 | 12,468 | 11,556 |
| 6 | 25,868 | 24,143 | 22,419 | 21,926 | 20,201 | 18,723 |
| 7 | 2,649 | 2,472 | 2,295 | 2,245 | 2,068 | 1,917 |
| 8 | 3,614 | 3,373 | 3,132 | 3,063 | 2,822 | 2,616 |
| 9 | 30,183 | 28,171 | 26,159 | 25,584 | 23,572 | 21,847 |
| 10 | 40,464 | 37,767 | 35,069 | 34,298 | 31,601 | 29,288 |
| 11 | 11,823 | 11,035 | 10,246 | 10,021 | 9,233 | 8,558 |
| Total Watershed | 294,806 | 275,153 | 255,499 | 249,884 | 230,230 | 213,384 |



**Table 5. Assigned debris flow damage classes**

| Damage Class | Description |
| --- | --- |
| 0 | Scour and transportation zones. Buildings assumed not present. Damage class 0 is blue on the Damage maps in Appendix B |
| 1 | Debris flow runout depth < 0.5 m |
| 2 | Debris flow depth generally between 0.5 and 1.5 m |
| 3 | Debris flow depth generally between 1.5 and 2.5 m |
| 4 | Debris flow depth generally > 2.5 m |