# Peer review of "LABS: an agent-based run-out program for shallow landslides"

_Natural Hazards and Earth System Sciences, 2020_

## Referee Comment (RC1) · Ivan Marchesini (Referee) · 25 Oct 2020

GENERAL COMMENTS

The manuscript describes a completely new agent-based software, LABS, devoted to modeling of debris flow runout, erosion e deposition at regional scale and given a limited amount of information about boundary conditions and physical parameters. In the first part of the manuscript the authors provide a rapid introduction to the software. Then they describe two application of the code: in Indonesia and Canada. Finally they conclude discussing the code and its limitations.

The manuscript is interesting and within the scope of NHESS even if I think that its scientific significance, quality and its presentation quality can be improved, also to

better support the interpretations and the conclusions derived from the results.

About the structure of the paper I note that frequently in the text (e.g. page 5, line 13 and line 23, or page 6 line 6 or page 8 lines 16 to 23 and others) there are comments and interpretations that should be reserved for the discussion or the conclusions. I also suggest to split the discussion of the model (and its limitations and performance) from the discussion of the results in the two study areas and to move there the text and concepts of sections 3.2.4 and 3.3.5.

The manuscript includes many figures (27) and possibly some of then can be further combined and or excluded or also improved (screenshots by themselves are not very informative). I think that legend is important in the figures showing the erosion and deposition cells.

Authors state that the software will be freely available for to no-profit groups. I wonder why not to release the code under an open source license (GPL as an example) in order to facilitate the re-usage and improvement, of this interesting tool, by the scientific community. I also wonder if the tool will be provided as a binary code for multiple OS (Win, MacOS, GNU/Linux)

SPECIFIC COMMENTS

Among the different papers dealing with regional modeling of debris flow runout cited in the introduction I would suggest to introduce these other two papers that are exactly dealing with the topic: • Mergili, M., Krenn, J., Chu, H.-J. (2015): r.randomwalk v1, a multi-functional conceptual tool for mass movement routing. Geoscientific Model Development 8: 4027-4043. doi:10.5194/gmd-8-4027-2015 • Mergili, M., Chu, H.-J. (2015): Integrated statistical modelling of spatial landslide probability. Natural Hazards and Earth System Sciences Discussions 3: 5677-5715. doi:10.5194/nhessd-3-5677-2015

In the description of the program it is said that the software uses 5m resolution DEM.

Is it not possible to run the tool on a grid having different size? If not I think that some comments on the memory requirement for running the tool on a given portion of the territory should be provided since this can hamper to use of the tool, on normal laptop or workstation, for modeling large areas.

Description of the program is rather short and not very detailed. As the tool is proposed to the scientific community I suggest to enlarge the description of the algorithm. As an example section 2.4 about spread is not very clear to me and the 3D Gaussian surface of figure 2 is not very informative. Moreover I'm not really sure that all the possible parameters visible in figure 20 (fan maximum slope, etc...) are fully described in section 2. I also suggest to clear if the model, after the run, alter the DEM, carving or uplifting it in correspondence of erosion and deposition areas. This would be an important tool since, when a an agent passes through a given cell, it could find the the DEM altered by an antecedent agent coming from another source and this could have an effect on the propagation of the second one. It seems the DEM is modified, from what we can read at lines 10-11 at page 6, but probably it should be made more clear.

Some concepts are repeated. As an example page 5/6 lines 26-27/1-2 or page 7 lines 10-11

I suggest to add information to the background for the first case study. As an example it is relevant to know the size of the study area, which DEM was used and if the DEM is pre- or post-event.

Section 3.2.2 is about calibration. However it is not clear how the model is calibrated. I would have expected that some of the parameters used by the model would have been changed to make the model match with the observed data but it seems to me that there is not such type of action. Why didn't the authors tried to tailor the model results to the ground truth? This has also to do with the model sensitivity. I really suggest to better discuss this point.

Lines 1-5 at page 7 describes figure 10 and rainfall triggered landslides and storm return period. I think that the chart should contain the known data in order to understand how they are fit by the curve and how many point were used to build such a relationship.

In section 3.2.3 it is said that random source location are placed based on a susceptibility map. It is relevant to describe how this susceptibility map was generated. Authors also mention existing terrain polygons but, if I'm not wrong, they were not mentioned before and I don't know what they are. It is not clear to me what they are. In the same section lines from 1 to 8 are not very clear to me. Probably experiment settings should be better explained. Again, in the same section, at line 10, authors say "once the historical event were calibrated.." but calibration phase was in section 3.2.2 and I'm not sure they are talking of that part of the manuscript but rather about the six storms.

At lines 14-16 of page 9 authors discuss a sort of susceptibility map. The citation is Palmer (2018) but the reference is "Palmer: Lake Cowichan and Youbou Slope Hazard Assessment, 2018." that I wasn't able to find.

At line 20 of page 9 authors say "The model was calibrated by simulating landslides within the study area, comparing the results to mapped and expected landslide behavior..". I think they can improve the description of what they intend with "mapped and expected landslide behavior".

At line 24 of page 9 authors say "magnitude frequency curves that are similar to other coastal BC data sets (Figure 18) with a similar rollover and distributions" but, taken the same area value, the model data differ till 1 order of magnitude of cumulated probability with the other curves. It seems, as a consequence, that area distribution is underestimated. Why non to try to better calibrate the model parameters to improve the matching of the model outputs with the frequency size distribution of the real inventories?

In section 3.3.2 manuscript declares that "Landslide initiation locations were created by importing randomly distributed points, a uniform distribution of points, and manually in the GIS tool within LABS". However there is no discussion about the effect of these different methods used to define the landslides initiation locations on the performance

of the models. There is only a rapid comment on section 3.3.3. On section 3.3.3 authors start with: "Once tested..". Are they meaning "Once calibrated.."?

At line 9 on page 10 there is the following text: "(both random and manually selected)" . But just a paragraph above authors state: "A user-based initiation-point selection method was used for the final model runs as this method generally resulted in landslide generation somewhat more frequently than randomly or uniformly generated points that would sometimes occur on a flatter portion of the slope". Sorry but I don't understand which is the method used, at the end.

Comment on lines 12-14 of page 10 are interesting but I wonder if they can be considered conservative given the fact that the magnitude frequency curve of the modeled landslides resulted in smaller landslides respect to those observed in other similar zones.

In section 4.3 there is a discussion about DEM resolution, that is fixed to 5m. As I said, I think that this should go together with an analysis of the memory requirement for running the tool. Having a fixed value for the resolution there are probably limitations (memory) about the maximum size of the area that can be studied using a computer.

TECHNICAL CORRECTIONS

Page 8, line 29: "700 mm" and "6000 mm" I suppose.. Is the -1 an error?

Page 10, line 29: are authors meaning Figure 24 and 25?

page 12 line 6: please remove "is"

---

## Referee Comment (RC2) · Anonymous Referee #2 · 26 Nov 2020

[referee-annotated manuscript omitted]

---

## Author Comment (AC1) · 25 Jan 2021

The authors are grateful to the reviewer for his constructive comments. We'll endeavor to respond appropriately below. A more fulsome response with figures is attached as a supplemental PDF. In our response below, reviewer comments are in square brackets, followed by our response outside of brackets.

[The manuscript includes many figures (27) and possibly some of then can be further combined and or excluded or also improved (screenshots by themselves are not very informative). I think that legend is important in the figures showing the erosion and deposition cells.]

We agree that some of the Figures can be combined and will do so on the revised

manuscript. We'll remove Figure 2 and replace it instead with the underlying algorithm.

[Authors state that the software will be freely available for to no-profit groups. I wonder why not to release the code under an open source license (GPL as an example) in order to facilitate the re-usage and improvement, of this interesting tool, by the scientific community. I also wonder if the tool will be provided as a binary code for multiple OS (Win, MacOS, GNU/Linux)]

Currently, we are not intending to release the code under an open source license. We are extremely interested in working with the larger scientific community and will freely provide the software for non-commercial use. The tool was written for the Windows platform and will not be provide as a binary code for multiple OS.

[Among the different papers dealing with regional modeling of debris flow runout cited in the introduction I would suggest to introduce these other two papers that are exactly dealing with the topic:]

Thank you for the references, we will include them in our update. The papers by Mergili and others take a decidedly different (though innovative) approach to landslide runout. We expect, based on our review, that they require additional software (UNIX, GRASS GIS) that are less common outside of academia, as well as calibration of the break criteria. While LABS also expects calibration by an expert, the methods are, in our opinion, more accessible to the user.

[In the description of the program it is said that the software uses 5m resolution DEM. Is it not possible to run the tool on a grid having different size? If not I think that some comments on the memory requirement for running the tool on a given portion of the territory should be provided since this can hamper to use of the tool, on normal laptop or workstation, for modeling large areas.]

The program is optimized for a 5 m DEM. Smaller or larger DEMs are likely to produce incorrect results. Comments about computer power and memory are almost certain

to be obsolete at time of (or shortly after) publication. We are currently running the program on a variety of laptops and limiting processing time by breaking down the DEM to reasonable sizes (currently several hundred km2). For slower computers there is a Region of Interest function within the program that allows the user to define an limited area for analysis.

[Description of the program is rather short and not very detailed. As the tool is proposed to the scientific community I suggest to enlarge the description of the algorithm. As an example section 2.4 about spread is not very clear to me and the 3D Gaussian surface of figure 2 is not very informative.]

Agreed. We will replace Figure 2 with a more detailed explanation of the spread mechanism. Please see the PDF for a better explanation (contains equations that aren't easily reproduced here).

[I also suggest to clear if the model, after the run, alter the DEM, carving or uplifting it in correspondence of erosion and deposition areas. This would be an important tool since, when a an agent passes through a given cell, it could find the the DEM altered by an antecedent agent coming from another source and this could have an effect on the propagation of the second one. It seems the DEM is modified, from what we can read at lines 10-11 at page 6, but probably it should be made more clear.]

You are correct, the model alters the DEM as it propagates and this in turn affects subsequent landslides. However, once the run is reset, the DEM is reverted to it's imported (original) form. This ensures that multiple runs are not affected by previous runs.

[Some concepts are repeated. As an example page 5/6 lines 26-27/1-2 or page 7 lines 10-11]

Right. We'll review and remove unnecessary repetition.

[I suggest to add information to the background for the first case study. As an example

it is relevant to know the size of the study area, which DEM was used and if the DEM is pre- or post-event.]

Great points. We'll provide the additional detail. For the record, the study area is 21.4 km2, relief is 2,646 m, the DEM is 2018, and it is post event.

[Section 3.2.2 is about calibration. However it is not clear how the model is calibrated. I would have expected that some of the parameters used by the model would have been changed to make the model match with the observed data but it seems to me that there is not such type of action. Why didn't the authors tried to tailor the model results to the ground truth? This has also to do with the model sensitivity. I really suggest to better discuss this point.]

We agree that this section needs additional explanation. Model calibration is completed iteratively using the controls within the program. The landslide professional runs the model per Section 3.2.2 and compares the results to mapped or historical landslides and ground-based evidence for travel distance, scour and deposition. Several methods may be employed including a visual comparison, magnitude-frequency comparison of mapped versus modeled landslides (e.g. Figure 13), and volume area relationships (e.g. Figure 7). The "Inspect" tool allows the user to examine the results pixel by pixel and the "One By One" advances individual agents through single time steps allowing for a much more detailed analysis of results. Typically, adjustments are made to the control sliders until better results are realized. This might require several runs. Control sliders adjust the shape and spread, and the volume eroded or deposited in each timestep. Note that the volume controls are new since the manuscript was submitted. Please see the supplemental PDF for a figure showing good vs. bad calibration using M-F analysis.

[In section 3.2.3 it is said that random source location are placed based on a suscepti­bility map. It is relevant to describe how this susceptibility map was generated. Authors also mention existing terrain polygons but, if I'm not wrong, they were not mentioned

before and I don't know what they are. It is not clear to me what they are. In the same section lines from 1 to 8 are not very clear to me. Probably experiment settings should be better explained. Again, in the same section, at line 10, authors say "once the historical event were calibrated.." but calibration phase was in section 3.2.2 and I'm not sure they are talking of that part of the manuscript but rather about the six storms.]

One of the biggest challenges we had writing this paper was to provide enough information to elicit interest in the model and show that it had wide application and robust performance while, at the same time, respecting typical article length. The susceptibility map was relevant in that it provided a logical place from which to initiate future landslides, however, methods of creating such maps are well understood and, in our opinion, unnecessary to repeat here. Suffice it to say that whatever the method of susceptibility mapping one uses, the landslide professional can then choose to initiate landslides from appropriate locations within the landscape. We will nonetheless seek to clarify the section in the updated manuscript. Line 10 should read, "Design floods were determined by bulking the debris flood model with sediment estimated for specific storm return periods from the calibrated debris flow model using LABS (Table 4)."

[At line 20 of page 9 authors say "The model was calibrated by simulating landslides within the study area, comparing the results to mapped and expected landslide behavior..". I think they can improve the description of what they intend with "mapped and expected landslide behavior".]

LABS is both predictive and probabilistic. We are not precisely recreating an existing or historic landslide, but instead trying to credibly produce predictions of landslides that may occur on the existing surface. We think that this is a strength of the program, however, it means that calibration includes a degree of expert judgement. The professional must decide whether modeled landslides travel along realistic paths, whether the paths are similar to those of historical events as mapped or as observable in the air photographs, whether the range of deposition and erosion approximates similar events in the same region, and finally, analytically, whether or not the magnitude frequency

characteristics are sufficiently similar to mapped characteristics, or justifiably different.

[At line 24 of page 9 authors say "magnitude frequency curves that are similar to other coastal BC data sets (Figure 18) with a similar rollover and distributions" but, taken the same area value, the model data differ till 1 order of magnitude of cumulated probability with the other curves. It seems, as a consequence, that area distribution is underestimated. Why non to try to better calibrate the model parameters to improve the matching of the model outputs with the frequency size distribution of the real inventories?]

We agree that this section could be clearer. We propose to change it to a more precise analysis: The tangent of the slope at a given probability of occurrence [See figure in PDF] was approximately equal for both modeled and mapped landslides. We thereby interpret that the model does a good job representing variability in landslide size distribution. However, mapped landslides generally occupied about twice the area of modeled landslides. Mapping is, in and of itself, a model. There are restrictions related to level of detail and a practical mapping scale. The mapper must make a choice between outlining landslides that are inferred to exist on steep slopes and precisely following the limited path visible among trees. In this case, the model appears to have better limited the landslide width to the actual path [New figure in PDF]. Mapped landslides include areas of steep gullies and slopes that are heavily forested after the identified event. We therefore interpret that the magnitudes of the mapped landslides are conservatively inflated and that is reflected in the curve in the figure in the supplemental PDF.

[In section 3.3.2 manuscript declares that "Landslide initiation locations were created by importing randomly distributed points, a uniform distribution of points, and manually in the GIS tool within LABS". However there is no discussion about the effect of these different methods used to define the landslides initiation locations on the performance of the models. There is only a rapid comment on section 3.3.3. On section 3.3.3 authors start with: "Once tested..". Are they meaning "Once calibrated.."?]

There was no difference in the results, except that some random/uniform points didn't

result in landslides (local slope was too flat). For landslides that initiated, the results were comparable.

[At line 9 on page 10 there is the following text: "(both random and manually selected)" . But just a paragraph above authors state: "A user-based initiation-point selection method was used for the final model runs as this method generally resulted in landslide generation somewhat more frequently than randomly or uniformly generated points that would sometimes occur on a flatter portion of the slope". Sorry but I don't understand which is the method used, at the end.]

Same as above. Using the random or uniform distribution of initiation points meant that some agents were generated on local slopes too flat to initiate a landslide response. Manual selection simply reduced the probability that this would occur.

[Comment on lines 12-14 of page 10 are interesting but I wonder if they can be considered conservative given the fact that the magnitude frequency curve of the modeled landslides resulted in smaller landslides respect to those observed in other similar zones.]

It's a fair question. Under current conditions, modeled landslides traveled consistently further than mapped landslides. Fanning behavior modeled did approximate vegetation changes on the fan, but exceeded what had been observed in the last several decades of air photograph interpretation. As per the discussion above, we interpret the difference in the reported magnitudes to be a result of mapping conservatism.

[In section 4.3 there is a discussion about DEM resolution, that is fixed to 5m. As I said, I think that this should go together with an analysis of the memory requirement for running the tool. Having a fixed value for the resolution there are probably limitations (memory) about the maximum size of the area that can be studied using a computer.]

Such a discussion will probably be outdated by the time this goes to press. Certainly by the following year. Computational demand is affected by the size of the area being

processed at a single time; however, LABS has a ROI button that allows the user to focus analysis on smaller portions of the DEM if required.

[Page 8, line 29: "700 mm" and "6000 mm" I suppose.. Is the -1 an error? Page 10, line 29: are authors meaning Figure 24 and 25? page 12 line 6: please remove "is"]

Page 8 Line 29 should read "Annual precipitation varies between 700 mm and 6,000 mm...". Page 10 does refer to Figures 24 and 25. Agree to remove "is" from Page 12.

Please also note the supplement to this comment:
https://nhess.copernicus.org/preprints/nhess-2020-233/nhess-2020-233-AC1-supplement.pdf

---

## Author Comment (AC2) · 26 Jan 2021

The authors are grateful to the anonymous reviewer. We will endeavor to respond appropriately below. Please refer to our supplementary PDF for additional information. In addressing these comments, we place the reviewer's original comment in square brackets, and our response below.

[In general, there is serious issue with respect to quantitative assessment of reliability of the modeling results in terms of calculated landslide volumes, depths and their run-out limits. All of these parameters are very useful and can be applied for practical hazard assessment, but the user needs to know how reliable the model outputs are. This is so far characterized mostly by qualitative, general statements.]

[Figure]

We respectfully disagree, but we certainly acknowledge that model reliability includes a high degree of expert judgement. The professional must decide whether modeled landslides travel along realistic paths, whether the paths are similar to those of historical events as mapped or as observable in the air photographs, whether the range of deposition and erosion approximates similar events in the same region, and finally, analytically, whether or not the magnitude frequency and area-volume characteristics are sufficiently similar to mapped characteristics, or justifiably different.

Because LABS is both predictive and probabilistic, it may not precisely recreate an existing or historic landslide, but instead tries to credibly produce predictions of landslides that may occur on the existing surface. It's also not a susceptibility model so we don't expect to conduct the type of reliability testing that we normally see (and need) for that type. Our best quantitative calibration tools are the M-F comparisons, and visual comparisons of landslide runout to mapped landslides and geomorphology (more below). We think that the predictive and probabilistic aspect of the program is a strength, and we include the ability to model many landslides to compare the range of responses between runs. As it happens, we have used detailed historic landslide studies to calibrate the current predictions, however, these are the subject of a different paper.

Model calibration is completed iteratively using the controls within the program. The landslide professional runs the model and compares the results to mapped or historical landslides and ground-based evidence for travel distance, scour and deposition. Several methods may be employed including a visual comparison, quantitative comparison of magnitude-frequency of mapped versus modeled results (please see the figure in the supplemental PDF) and volume-area relationships or simple landslide length comparisons.

The "Inspect" tool allows the user to examine the results (including depth) pixel by pixel and the "One By One" advances individual agents through single time steps allowing for a much more detailed analysis of results. These results can be compared to known ground investigations.

[Figure]

Typically, adjustments are made to the control sliders until better results are realized. This might require several runs. Control sliders adjust the shape and spread, and the volume eroded or deposited in each timestep. Note that the volume controls are new since the manuscript was submitted. We provide a figure in the supplemental PDF that shows the difference between a poorly calibrated result and a well calibrated result using M-F analysis.

[I also see serious problem with respect to maps you presented in the manuscript. The maps you show lack legends and geographic coordinates, which needs to be added to allow the user to get all information they show.]

Agreed. We will update the maps.

Author's Note: Additional marginal comments provided by the reviewer in a markup attachment were largely either editorial, or along the lines of the question above. The questions are generally reasonable, and the authors can certainly update the manuscript to clarify. A couple of comments/questions were unique and we are adding those below.

[Please add reference which would provide definition of "debris floods" as this would significantly contribute to better understanding of the topic.]

Agreed. We propose to introduce the recent Church and Jakob (2020) reference. Church, M, and M Jakob. 2020. "What is a debris flood?" Water Resources Research 17.

[Several comments related to quantifying observations instead of using qualitative adjectives].

Agreed. This is, by and large, a reasonable request and we will update the manuscript accordingly.

[This fact [that distal margins of landslides tend to be inundated less frequently than the main landslide body] along with characteristics mentioned in the preceding paragraph

can be serious limitation of the model if we would search and answer where it is safe to build houses with respect to the expected run-out. Could you quantify or describe in more quantitative manner the uncertainty related to the margins of the modeled run-out?]

We consider this to be a strength of the program. The fact that we can run a simulation multiple times and get what we believe is credible variation between runs allows the user to better estimate the potential footprint as in the following example. LABS allows you to show both the overall footprint and the most likely footprint for a specified topography (the current DEM). We provide an example of specific analysis to this effect in the supplemental PDF.

[Please explain . . . narrow shape of the transportation paths. It seems that some problems with DEM could be involved! Please, check it.]

The narrow linear shape of the transportation paths and the potential of DEM error is considered in the paper. It is almost certainly a limitation at the DEM scale, however, it is also consistent with the actual mapped landslides. We ran some additional analysis using only the mapped landslides and provide the results as a figure in the supplemental PDF. The modeled and the mapped landslides paths are the same.

[I think that [the runout] probability also largely depends on the initial volume of the material. Please consider this in your conclusions. It would be also nice if you may show calculations where the initial volume of landslide mass was larger.]

Runout does indeed depend on the initial volume, as well as the difference in available entrainment along the landslide path. The professional landslide specialist needs to consider these criteria when calibrating the model. The latest version of the model can increase or decrease the initial volumes, and the scour and deposition to match the geomorphologically interpreted criteria.

Please also note the supplement to this comment:

https://nhess.copernicus.org/preprints/nhess-2020-233/nhess-2020-233-AC2-supplement.pdf
* * *

---

## Author Response (AR1)

The authors are grateful to the reviewers for their constructive comments. We'll endeavor to respond appropriately below.

**Comments from 1st Reviewer**

[*The manuscript includes many figures (27) and possibly some of then can be further combined and or excluded or also improved (screenshots by themselves are not very informative). I think that legend is important in the figures showing the erosion and deposition cells.*]

Combined or removed figures as warranted. Added figures to explain some of the questions from the reviews. Net reduction in figures (to 24). Legends were added to some figures, but screenshots remain and some figures have descriptions in captions instead.

[*Authors state that the software will be freely available for to no-profit groups. I wonder why not to release the code under an open source license (GPL as an example) in order to facilitate the re-usage and improvement, of this interesting tool, by the scientific community. I also wonder if the tool will be provided as a binary code for multiple OS (Win, MacOS, GNU/Linux)*]

Currently, we are not intending to release the code under an open source license. We are extremely interested in working with the larger scientific community and will freely provide the software for non-commercial use. The tool was written for the Windows platform and will not be provide as a binary code for multiple OS.

[*Among the different papers dealing with regional modeling of debris flow runout cited in the introduction I would suggest to introduce these other two papers that are exactly dealing with the topic:*]

Thank you for the references, we included them in our update. The papers by Mergili and others take a decidedly different (though innovative) approach to landslide runout. We expect, based on our review, that they require additional software (UNIX, GRASS GIS) that are less common outside of academia, as well as calibration of the break criteria. While LABS also expects calibration by an expert, the methods are, in our opinion, more accessible to the user.

[*In the description of the program it is said that the software uses 5m resolution DEM. Is it not possible to run the tool on a grid having different size? If not I think that some comments on the memory requirement for running the tool on a given portion of the territory should be provided since this can hamper to use of the tool, on normal laptop or workstation, for modeling large areas.*]

The program is optimized for a 5 m DEM. Smaller or larger DEMs are likely to produce incorrect results. Comments about computer power and memory are almost certain to be obsolete at time of (or shortly after) publication. We are currently running the program on a variety of laptops and limiting processing time by breaking down the DEM to reasonable sizes (currently several hundred $km^2$). For slower computers there is a Region of Interest function within the program that allows the user to define an limited area for analysis.

[*Description of the program is rather short and not very detailed. As the tool is proposed to the scientific community I suggest to enlarge the description of the algorithm. As an example section 2.4 about spread is not very clear to me and the 3D Gaussian surface of figure 2 is not very informative.*]

Agreed. We replaced Figure 2 with a more detailed explanation of the spread mechanism.

[*I also suggest to clear if the model, after the run, alter the DEM, carving or uplifting it in correspondence of erosion and deposition areas. This would be an important tool since, when a an agent passes through*]

*a given cell, it could find the the DEM altered by an antecedent agent coming from another source and this could have an effect on the propagation of the second one. It seems the DEM is modified, from what we can read at lines 10-11 at page 6, but probably it should be made more clear.*]

You are correct, the model alters the DEM as it propagates and this in turn affects subsequent landslides. However, once the run is reset, the DEM is reverted to it's imported (original) form. This ensures that multiple runs are not affected by previous runs.

[*Some concepts are repeated. As an example page 5/6 lines 26-27/1-2 or page 7 lines 10-11*]

We have endeavored to remove unnecessary repetition.

[*I suggest to add information to the background for the first case study. As an example it is relevant to know the size of the study area, which DEM was used and if the DEM is pre- or post-event.*]

Done.

[*Section 3.2.2 is about calibration. However it is not clear how the model is calibrated. I would have expected that some of the parameters used by the model would have been changed to make the model match with the observed data but it seems to me that there is not such type of action. Why didn't the authors tried to tailor the model results to the ground truth? This has also to do with the model sensitivity. I really suggest to better discuss this point.*]

Clarified in manuscript.

[*In section 3.2.3 it is said that random source location are placed based on a susceptibility map. It is relevant to describe how this susceptibility map was generated. Authors also mention existing terrain polygons but, if I'm not wrong, they were not mentioned before and I don't know what they are. It is not clear to me what they are. In the same section lines from 1 to 8 are not very clear to me. Probably experiment settings should be better explained. Again, in the same section, at line 10, authors say "once the historical event were calibrated.." but calibration phase was in section 3.2.2 and I'm not sure they are talking of that part of the manuscript but rather about the six storms.*]

Clarified in manuscript.

[*At lines 14-16 of page 9 authors discuss a sort of susceptibility map. The citation is Palmer (2018) but the reference is "Palmer: Lake Cowichan and Youbou Slope Hazard Assessment, 2018." that I wasn't able to find.*]

The report is publicly available here: [Natural Hazard Risk Assessments | Cowichan Valley Regional District (cvrd.ca)](Natural Hazard Risk Assessments | Cowichan Valley Regional District (cvrd.ca))

[*At line 20 of page 9 authors say "The model was calibrated by simulating landslides within the study area, comparing the results to mapped and expected landslide behavior..". I think they can improve the description of what they intend with "mapped and expected landslide behavior".*]

Clarified in manuscript.

[*At line 24 of page 9 authors say "magnitude frequency curves that are similar to other coastal BC data sets (Figure 18) with a similar rollover and distributions" but, taken the same area value, the model data differ till 1 order of magnitude of cumulated probability with the other curves. It seems, as a consequence,*

*that area distribution is underestimated. Why non to try to better calibrate the model parameters to improve the matching of the model outputs with the frequency size distribution of the real inventories?*]

Updated and clarified section.

[*In section 3.3.2 manuscript declares that "Landslide initiation locations were created by importing randomly distributed points, a uniform distribution of points, and manually in the GIS tool within LABS". However there is no discussion about the effect of these different methods used to define the landslides initiation locations on the performance of the models. There is only a rapid comment on section 3.3.3. On section 3.3.3 authors start with: "Once tested..". Are they meaning "Once calibrated..?"*]

Clarified in manuscript.

[*At line 9 on page 10 there is the following text: "(both random and manually selected)" . But just a paragraph above authors state: "A user-based initiation-point selection method was used for the final model runs as this method generally resulted in landslide generation somewhat more frequently than randomly or uniformly generated points that would sometimes occur on a flatter portion of the slope". Sorry but I don't understand which is the method used, at the end.*]

Clarified in manuscript.

[*Comment on lines 12-14 of page 10 are interesting but I wonder if they can be considered conservative given the fact that the magnitude frequency curve of the modeled landslides resulted in smaller landslides respect to those observed in other similar zones.*]

Updated discussion in manuscript and provided M-F curve for mapped vs. modeled landslides of same slope.

[*In section 4.3 there is a discussion about DEM resolution, that is fixed to 5m. As I said, I think that this should go together with an analysis of the memory requirement for running the tool. Having a fixed value for the resolution there are probably limitations (memory) about the maximum size of the area that can be studied using a computer.*]

Such a discussion will probably be outdated by the time this goes to press. Certainly by the following year. Computational demand is affected by the size of the area being processed at a single time; however, LABS has a ROI button that allows the user to focus analysis on smaller portions of the DEM if required.

[*Page 8, line 29: "700 mm" and "6000 mm" I suppose.. Is the -1 an error? Page 10, line 29: are authors meaning Figure 24 and 25? page 12 line 6: please remove "is"*]

Done.

**Comments from the 2nd reviewer:**

[*In general, there is serious issue with respect to quantitative assessment of reliability of the modeling results in terms of calculated landslide volumes, depths and their run-out limits. All of these parameters are very useful and can be applied for practical hazard assessment, but the user needs to know how reliable the model outputs are. This is so far characterized mostly by qualitative, general statements.*]

We acknowledge that model reliability includes a high degree of expert judgement, however, we think that a landslide professional will have a relatively easy time calibrating the model. We've added clarity around calibration of the model in the manuscript.

[*Please add reference which would provide definition of "debris floods" as this would significantly contribute to better understanding of the topic*.]

Done

[*This fact* [that distal margins of landslides tend to be inundated less frequently than the main landslide body] *along with characteristics mentioned in the preceding paragraph can be serious limitation of the model if we would search and answer where it is safe to build houses with respect to the expected run-out. Could you quantify or describe in more quantitative manner the uncertainty related to the margins of the modeled run-out?*]

We consider this to be a strength of the program. The fact that we can run a simulation multiple times and get what we believe is credible variation between runs allows the user to better estimate the potential footprint as in the following example. LABS allows you to show both the overall footprint and the most likely footprint for a specified topography (the current DEM).

[*Please explain … narrow shape of the transportation paths. It seems that some problems with DEM could be involved! Please, check it.*]

The narrow shape of the transport paths is typical for debris flows in many areas of the world. It's not a limitation of the model but a reflection of the topography. That said, there is some potential for DEM error that is considered in the paper. It is almost certainly a limitation at the DEM scale, however, it is also consistent with the actual mapped landslides. (Figure attached).

[Figure]

Strong linear orientation of modeled landslides on the North Shore when hundreds of landslides are viewed at once (A). The results look more reasonable (though still linear) when compared to just the mapped landslides (B) and (C). Google Earth image in the background of (A).

[*I think that* [the runout] *probability also largely depends on the initial volume of the material. Please consider this in your conclusions. It would be also nice if you may show calculations where the initial volume of landslide mass was larger*.]

Runout does indeed depend on the initial volume, as well as the difference in available entrainment along the landslide path. The professional landslide specialist needs to consider these criteria when calibrating the model. The latest version of the model can increase or decrease the initial volumes, and the scour and deposition to match the geomorphologically interpreted criteria.